# Volatile Organic Compounds and Ozone in Rocky Mountain National Park during FRAPPÉ

Katherine B. Benedict[1], Yong Zhou[1], Barkley C Sive[2], Anthony J. Prenni[2], Kristi A. Gebhart[3], Emily V. Fischer[1], Ashley Evanoski-Cole[1], Amy P. Sullivan[1], Sara Callahan[1], Bret A. Schichtel[3], Huiting Mao[4], Ying Zhou[4], Jeffrey L. Collett, Jr. [1]

[1]Department of Atmospheric Science, Colorado State University, Fort Collins, CO 80523, USA
[2]National Park Service, Air Resources Division, Lakewood, CO 80225, USA
[3]National Park Service, Air Resources Division, Fort Collins, CO 80523, USA
[4]Department of Chemistry, State University of New York College of Environmental Science and Forestry, Syracuse, NY, 13210, USA

*Correspondence to*: Katherine B. Benedict (katherine.benedict@colostate.edu)

**Abstract.** The 2014 Front Range Air Pollution and Photochemistry Éxperiment (FRAPPÉ) aimed to better characterize summertime air quality in the Northern Front Range Metropolitan Area (NFRMA) and its impact on surrounding areas. As part of this study, measurements of gas and particle phase species were collected in Rocky Mountain National Park (ROMO), located in the mountains west of the urban North Front Range corridor from July – October 2014. We report on measurements of ozone from two locations in the park and a suite of volatile organic compounds (VOCs) measured using a continuous real-time gas chromatography system and a quadrupole proton-transfer reaction-mass spectrometer at the ROMO Longs Peak air quality site. We also measured VOCs using canister samples collected along transects connecting the NFRMA and ROMO. These datasets show that ROMO is impacted by NFRMA emission sources, and high observed mixing ratios of VOCs associated with oil and gas extraction (e.g. ethane) and urban sources (e.g. ethene and $C_2Cl_4$) occur during periods of upslope transport. Hourly ozone mixing ratios exceeded 70 ppb during six events. Two of the six events were largely associated with VOCs from the oil and gas sector, three high ozone events were associated with a mixture of VOCs from urban and oil and gas sources, and one high ozone event was driven by a stratospheric intrusion. For the high ozone events most associated with emissions from oil and gas activities, we estimate that VOCs and $NO_x$ from sources along the Front Range contributed ~20 ppbv of additional ozone.

## 1. Introduction

Ozone ($O_3$) is a regulated pollutant that can impact both human and ecosystem health. Monthly mean daytime $O_3$ has been increasing in the Rocky Mountains and throughout much of the west (Strode et al., 2015), and there are concerns about $O_3$ levels and exceedances of the $O_3$ National Ambient Air Quality Standard (NAAQS) at high elevation sites (Christensen et

al., 2015; Musselman and Korfmacher, 2014). In Rocky Mountain National Park (ROMO), a protected Class I area with millions of visitors each year (National Park Service, 2017), $O_3$ mixing ratios often exceed the current NAAQS standard of 70 ppb and vegetation injury thresholds (Kohut et al., 2012). A timeline of $O_3$ mixing ratios measured at the ROMO Longs Peak (ROMO-LP) CASTNET (Clean Air Status and Trends NETwork) site from January 2010 to December 2015 is shown in Figure

5    1. In general, $O_3$ mixing ratios were lower during the summer of 2014 than in the preceding four years; specifically, the peak $O_3$ mixing ratios were lower and there were fewer hours when mixing ratios exceeded 70 ppb. Weather conditions, including greater rainfall and cloud cover than is typical during the summer months, contributed to lower $O_3$ levels in ROMO and on the Front Range (Cheadle et al., 2017; McDuffie et al., 2016). In addition to transport and $O_3$ production from regional source emissions, the U.S. Intermountain West (all elevations) has a higher probability of stratospheric $O_3$ reaching the troposphere

10    from tropopause folding (related to mid-latitude jet or cutoff low) and stratospheric intrusion events (Lin et al., 2015; Musselman and Korfmacher, 2014; Reddy and Pfister, 2016; Wooldridge et al., 1997). At Great Basin National Park, another high elevation park in the west, elevated $O_3$ mixing ratios have been shown to be impacted by long-range transport, regional transport, and high-altitude transport (Christensen et al., 2015), highlighting the complex nature of $O_3$ sources at these Intermountain West sites.

15       ROMO is in close proximity to the North Front Range Metropolitan Area (NFRMA), which includes Denver, Boulder, and Fort Collins (urban areas) (Figure 2), in addition to the Denver-Julesburg Basin, which has seen significant increases in oil and natural gas production. For example, in nearby Weld County, the number of active wells doubled between January 2008 and July 2015, to over 27,000 [Colorado Oil and Gas Conservation Commission (COGCC) 2/2016]. Previous studies have extensively characterized NFRMA volatile organic compound (VOC) emissions, including those from oil and

20    natural gas extraction activities, and their influence on air quality at the Boulder Atmospheric Observatory (Gilman et al., 2013; McDuffie et al., 2016; Pétron et al., 2012; Swarthout et al., 2013). These studies demonstrate the extremely high mixing ratios of alkanes in the NFRMA, with maximum observed ethane and propane mixing ratios exceeding 100 ppbv (Swarthout et al., 2013) compared to mean values of 0.56-8.74 ppbv and 0.29-6.05 ppbv, respectively, in urban areas (Baker et al., 2008). The extent that these emissions reach ROMO has not been previously determined, although previous studies have shown that

agricultural ammonia emissions from a similar geographic source region are frequently transported into ROMO (Beem et al., 2010; Benedict et al., 2013).

The Front Range Air Pollution and Photochemistry Experiment (FRAPPÉ) took place in the summer of 2014 and was focused on characterizing summertime air quality and $O_3$ formation in the NFRMA.  FRAPPÉ included extensive ground, mobile and aircraft measurements throughout the region, including ROMO. This work focuses on a primary goal of the measurements at ROMO – to better understand the factors contributing to high $O_3$ in the park including potential contributions of the NFRMA.  These data are meant to inform strategies to reduce $O_3$ in the park and the exposure of visitors and ecosystems to elevated $O_3$ levels.  Measurements at the ROMO Longs Peak site (40.2783°N, 105.5457°W; 2784 m ASL) during FRAPPÉ included a suite of VOCs in addition to $O_3$ and meteorological parameters, which are routinely measured at the site.  Whole air canister samples were also collected along U.S. Route 34 from the NFRMA to higher elevation sites (3600 m) to examine spatial variations in VOC concentrations (Figure 2).  Ozone and meteorological measurements were also made at an additional high elevation site within ROMO, located off Trail Ridge Road (ROMO-TR; 40.3900° N, 105.6865° W, 3498 m ASL).  Additional measurements carried out at ROMO during FRAPPÉ are the focus of separate publications (Benedict et al., 2018; Zaragoza et al., 2017).

## 2. Methods

### 2.1 In situ Volatile Organic Compound Measurements

A cryogen-free concentration system coupled with two gas chromatographs (GC) was installed in the Colorado State University Mobile Laboratory for near real-time (1-hr sample frequency) analysis of VOCs at the Longs Peak Monitoring Station at ROMO from 17 July to 4 September 2014.  Ambient air was drawn from 5.25 m above ground level using a 3 m × 0.635 cm outer diameter (o.d.) stainless steel line followed by a 3 m × 0.4 cm inner diameter (i.d.) PFA Teflon line. Air flow was maintained continuously at ~7 L min$^{-1}$ using a metal bellows pump (MB-158, Senior Flexionics, Sharon, MA) and delivered a pressure of 10 psig. The inlet residence time was <1 s.  A sub-stream of the pressurized air (~2.5 L min$^{-1}$) was directed to a water management system, which reduced water vapour before entering the analytical system. The water management system consisted of a 7.5 m × 0.635 cm o.d. stainless steel coil and a stainless steel liquid water reservoir, which

was housed in a commercial refrigerator and set at ~2° C.  The total volume of the system was ~380 ml, yielding a residence time of <10 s. The VOC analytical system was similar to those used in previous studies (Abeleira et al., 2017; Sive et al., 2005) and utilized a Qdrive 2s102K cryocooler (Chart, Inc., Troy, NY) that was capable of cooling a sample enrichment loop from 100 °C to -180 °C in 25 minutes for sample concentration.  Additional details are provided in the supplemental information.

Two different whole air standards were alternately analysed every 10 samples throughout the campaign for quantification of the target gases as well as to monitor the system's performance; the standard mixing ratios were representative of clean and polluted levels encountered at ROMO. The measurement precision, represented by the relative standard deviation (RSD) of the peak areas for each compound in the standards, was 1-8% for the non-methane hydrocarbons (NMHCs), 3-11% for halocarbons, and 3-5% for alkyl nitrates. A list of VOCs measured during this study using the *in situ* GC is shown in Table 1.

**2.2.1 Canister sampling**

Whole air canister samples were collected at eight non-urban locations during the campaign to investigate the spatial distributions of VOCs.  Canister samples were collected in the Big Thompson Canyon along Route 34 en route to the ROMO Longs Peak site three days a week during the campaign; several transects were also continued up Trail Ridge Road to the continental divide at the Alpine Visitors Center in ROMO (ROMO-TR, Figure 2) when upslope transported was expected.

Canisters were typically collected between 13:00 – 18:00 (local time). A total of 39 individual VOCs were quantified from the canister samples in our laboratory at Colorado State University using a five-channel, three-GC analytical system, which employed three FIDs, one ECD and one mass spectrometer (MS). The gases analysed included $C_2$-$C_{10}$ NMHCs, $C_1$-$C_2$ halocarbons, $C_1$-$C_5$ alkyl nitrates, selected reduced sulphur compounds and OVOCs (oxygenated VOCs).  A complete list of VOCs measured from the canister samples using the lab analytical system is included in Table 1.

The analytical system and methodology used for this study are similar to those used in previous studies (Russo et al., 2010b; Sive, 1998; Zhou et al., 2010).  Multiple standards were used during sample analysis (analysed every 10 samples). The whole air working standards employed for this work have mixing ratios representative of clean free tropospheric air (NMHCs mixing ratios on the order of ~10-100 pptv) and suburban air (NMHC mixing ratios on the order of ~1-2 ppbv), thus bracketing the low and high ranges for the measurements at ROMO (e.g. Russo et al., 2010a). The measurement precision, represented

by the relative standard deviation (RSD) of the peak areas for each compound in the whole air standards, was 1-8% for the NMHCs, 3-10% for halocarbons, and 3-5% for the alkyl nitrates.

### 2.2.2 Proton Transfer Reaction-Mass Spectrometer (PTR-MS)

Selected VOCs and OVOCs were monitored in real-time by a quadrupole proton-transfer reaction-mass spectrometer located in the NPS air quality monitoring shelter (PTR-MS; Ionicon Analytik, Innsbruck, Austria) from 14 July to 8 September 2014. The PTR-MS sampled air that was continuously drawn through a 13 m $\times$ 1.27 cm o.d. (0.9525 cm i.d.) PFA Teflon inlet from 10 m above ground level. The flow rate through the sample line was ~70 L min$^{-1}$, resulting in a <1 s residence time. A diaphragm pump was used to draw a sub-stream of air off the main inlet line through a 1 m $\times$ 0.635 cm PFA Teflon line at a flow rate of 1 L min$^{-1}$ from which the PTR-MS sampled.

The PTR-MS was operated with a drift tube pressure and temperature of 2.0 mbar and 45 °C, respectively, and a potential of 600 V applied over the length (9.6 cm) of the drift tube. The ion source water flow rate was 11 ml min$^{-1}$ and the discharge current was 8 mA. A series of 30 masses was monitored continuously; six masses were monitored for diagnostic purposes while the remaining 24 masses corresponded to VOCs of interest. The dwell time for each of these 24 masses ranged from 1-10 s, yielding a total measurement cycle of ~3 min. The system was zeroed every 25 hours for 10 cycles by diverting the flow of ambient air through a heated catalytic converter (0.5% Pd on alumina at 550 °C) to determine system background signals. Calibrations for the PTR-MS system were conducted using three different high-pressure cylinders containing synthetic blends of selected NMHCs and OVOCs at ppbv levels. Each of the cylinders used in the calibrations had an absolute accuracy of <±5% for all gases. Standards were diluted to atmospheric mixing ratios (ppbv to pptv levels) with catalytic converter-prepared zero air adjusted to maintain the humidity of the sampled air for 6-10 cycles every 25-75 hours. Mixing ratios for each gas were calculated by using the normalized counts per second which were obtained by subtracting out the non-zero background signal for each compound. The PTR-MS precision was estimated from counting statistics and ranged from 6-15%.

**2.3 Ozone measurement at Longs Peak and Trail Ridge Road**

**2.3.1 Trail Ridge Road**

The third generation of a custom built portable $O_3$ monitoring system (POMS3, Air Resource Specialists, Inc., Fort Collins, CO) was deployed at a high elevation site located off of the Ute Trail in ROMO (40.3900˚ N, 105.6865˚ W, 3498 m ASL)

during FRAPPÉ for $O_3$ and meteorological measurements.  Measurements were conducted from 11 July – 18 September 2014. The system incorporated two 2B Technologies $O_3$ analysers, a 6.1 m sampling mast, a data acquisition system, and meteorological instruments for measurements of wind speed, wind direction, temperature, relative humidity, solar radiation, and precipitation.  The POMS3 unit was fitted with solar panels and batteries sufficient for operation during the campaign. Hourly averages of $O_3$ and meteorological parameters were reported for this site. More information on the Trail Ridge $O_3$

measurements and calibration procedures can be found in the supplement.

**2.3.2 Longs Peak**

Ozone and meteorological measurements have been measured continuously at the ROMO Longs Peak site since 1987, as part of the National Park Service Gaseous Pollutant Monitoring Program.  As a regulatory $O_3$ monitoring station, the

sampling methods for gaseous and meteorological monitoring are based on the 40 CFR Part 58 requirements.  The Longs Peak site includes a temperature-controlled shelter that houses the $O_3$ analyser and calibration system. The $O_3$ inlet and meteorological measurements are located on top of a 10-m tower. Ozone is measured by UV absorption utilizing a Thermo 49i $O_3$ analyzer coupled with a Thermo 49i $O_3$ reference station for nightly zero, precision and span checks. One minute $O_3$ data are recorded and hourly averages are generated from the 1-minute $O_3$ data.  The suite of meteorological measurements

include wind speed, wind direction, relative humidity, temperature, solar radiation and precipitation, and 1-minute meteorological data were acquired during FRAPPÉ.  As with the POMS3 site, further details can be found in the Gaseous Pollutant Monitoring Program 2015 Quality Assurance Project Plan (https://ard-request.air-resource.com/Project/documents.aspx).

**2.4 Data filtering to remove local source effects**

Initial analysis of FRAPPÉ data showed unexpectedly high propane concentrations relative to other VOCs at the ROMO-LP site. The highest propane concentrations were associated with the lowest wind speeds, suggesting that a leaking propane tank used by a nearby youth camp (<200 m) was a local source. Typically in the U.S., propane fuel is primarily composed of propane (at least 90%), with no more than 2.5% butanes and maximum 5% propylene (ASTM Standard D1835-16, 2016). To better understand the influence of regional VOC emissions and transport on the distributions of VOCs and $O_3$ in ROMO, it was necessary to filter out alkane, alkene, and BTEX (benzene, toluene, ethylbenzene, xylene) data that were affected by the local emissions. In this study, data with ethane/propane ratios <1 were considered to be propane tank influenced and were removed. The resulting ratio of ethane to propane for filtered data (1.2) was similar to that determined for transect samples (1.13) taken at multiple locations in ROMO at different times, supporting the data-filtering approach.

**3. Temporal and Spatial Trends in Volatile Organic Compounds**

Throughout the campaign, hourly VOC mixing ratios varied by 1 to 3 orders of magnitude (Table 1). Figure 3 and Figure S1 show this variability for several VOCs as well as for $O_3$ at the ROMO-LP and ROMO-TR sites from 17 July - 4 September, 2014. The $C_2$-$C_5$ alkanes, $C_2$-$C_4$ alkenes, ethyne, aromatics (benzene and toluene), $C_2$-$C_5$ alkyl nitrates, tetrachloroethylene ($C_2Cl_4$), trichloroethylene ($C_2HCl_3$) and OVOCs, such as methanol, generally had similar temporal variations during most of the study period. The similar temporal patterns suggest that many of the concentration changes are driven by meteorology, with westerly flow generally transporting cleaner air, and easterly flow transporting pollutants from the NFRMA. However, in some cases, differences between species' mixing ratio enhancements likely reflect specific sources influencing air masses that reach the park.

All of the VOCs from Figure 3 have sources in the NFRMA. For example, light alkanes (e.g. ethane) have oil and natural gas sources in northeastern Colorado and have been observed in high abundance along the Front Range (Abeleira et al., 2017; Gilman et al., 2013; Swarthout et al., 2013). Vehicle combustion, prevalent throughout the NFRMA, is an important source of light alkenes, ethyne, and aromatics (e.g. benzene) (Koppmann, 2007), although oil and gas emissions have also been shown to be the most important source of aromatics, specifically benzene, at certain locations (Abeleira et al., 2017; Halliday et al., 2016). $C_2Cl_4$ and $C_2HCl_3$ are widely used as solvents and dry cleaning fluids and are frequently used as tracers

of urban air masses (Blake et al., 1996). Methanol has vegetative sources and potential oil and gas sources in some basins where it is used during cold weather to inhibit equipment freezing (Warneke et al., 2014), but, more importantly for the Front Range, in summer this trace gas also comes from animal agriculture and confined animal feeding operations (CAFOs)(Sun et al., 2008). Given the co-location of agricultural and oil and gas related sources in NE Colorado, it is difficult to ascertain the

magnitude of each source, but it is likely that agriculture and CAFOs would be the larger source in the NFRMA and surrounding areas during the summer.

To better understand VOC influence from these sources, air mass transport patterns were examined with back trajectories started hourly at 10 m above ground with previous air mass positions traced backward in time for two days using the Hybrid Single-Particle Lagrangian Integrated Trajectory (HYSPLIT) model ver 4.9 (Stein et al., 2015). Input data are from

the North American Model with a grid resolution of 12 km (NAM12) (Janjic, 2003). Back trajectory residence time analyses are a long-established method to examine upwind air mass transport pathways for periods of interest (Ashbaugh et al., 1985). Previous back trajectory analyses for ROMO captured most meso-synoptic scale patterns, but may underestimate the easterly upslope flow into the park (Gebhart et al., 2011, 2014). In Figure 4a, the Overall Residence Time (ORT) shows the relative probability for transport from the areas upwind of ROMO-LP during all hours of the study without regard to observed air mass

composition. Results indicate that during the study period, air masses arrived from all directions, but areas most frequently upwind were areas to the southwest and the Front Range. Similarly, High Mixing Ratio Residence Times (HRT) were generated using only hours when a measured VOC mixing ratio at ROMO-LP exceeded the 90th percentile of study values. The HRT for ethane, which primarily is from leakage during the production and transport of natural gas (Tzompa-Sosa et al., 2017), is shown in Figure 4b. A sparser pattern results from including only 10% of the hours. Many of the same areas are

upwind as in the ORT, but areas to the east have a higher probability of being upwind during this subset of high concentration time periods.

Figure 5 presents differences between transport patterns during high mixing ratio measurements (exceeding the 90th percentile), as compared to all study conditions. The difference maps in Figure 5 are calculated by subtracting the overall probability of transport (ORT) from the probability of transport on a high mixing ratio hour (HRT), or HRT-ORT (Poirot et

al., 2001) . Figure 5a shows the results for ethane (Figure 4b – Figure 4a). This more clearly shows that high concentrations

of ethane are preferentially associated with transport from northeastern Colorado, where large oil and gas emissions occur (Figure 2). Figure 5b shows a similar analysis, but for $C_2Cl_4$ (urban tracer) measurements. The results again point to the NFRMA, but with more of an influence from further south toward Denver because of its widespread usage in the region. Figure 5c shows results for isoprene in this case, wooded mountainous areas to the west are upwind more often than average.

Plots for ethene (tracer for fuel combustion) (Figure 5d) and ethyne (not shown) also point to the NFRMA. These results are not surprising, as upslope conditions from the east are expected to transport pollutants to the park. However, this is the first time that *in situ* VOC tracers have been used to demonstrate the many Front Range sources that impact ROMO.

At their peak values, NMHC mixing ratios observed at ROMO can be of comparable magnitude to urban/industrial regions, further suggesting a significant impact of the polluted NFRMA on ROMO during specific events. The range of mixing

ratios observed at ROMO overlaps the range of observations compiled by Baker et al. (2008) from 28 cities across the United States. Although measurements at ROMO were lower than at most other locations shown in Figure 6, mixing ratios of these VOCs are much higher at ROMO than is typical for remote sites (Jobson et al., 1994; Rindsland et al., 2002; Rudolph, 1995; Simpson et al., 2012). Observed mixing ratios during afternoons with upslope flow can be similar to those observed simultaneously in the Front Range (see maximum values in Table 1). ROMO is a Class I area that is afforded the highest level

of air quality protection, while these other locations are in major source regions. The fact that maximum NMHC levels at ROMO are of similar magnitude to these other areas suggests a significant impact on the park from anthropogenic emissions, at least during episodic transport. Figure 6 also includes measurements made during FRAPPÉ at the Platteville Atmospheric Observatory (PAO), a rural location surrounded by agriculture and intensive oil and gas development in northeastern Colorado (Halliday et al., 2016). Observed mixing ratios at this site tended to be the highest of those included in this figure. With the

exception of benzene and ethyne, measured mixing ratios from the transect samples (gray) and other measurements made in the Front Range (Abeleira et al., 2017; McDuffie et al., 2016; Swarthout et al., 2013) generally exceed those measured in ROMO. The transect samples were collected next to the road and only during daytime hours, when upslope flow from the east is prevalent. Both factors likely contribute to the larger values measured compared to the measurements at ROMO. The ROMO and transect ethyne mixing ratios presented in this figure likely represent background as there is significant overlap

for all sites with the range of means from 28 U.S. cities. Benzene mixing ratios also suggest background values for the ROMO

and transect samples while slightly elevated mixing ratios at BAO and PAO indicate additional sources, likely from oil and gas (Halliday et al., 2016).

The ratio of i-pentane to n-pentane has been used to identify air masses impacted by oil and gas emissions. Although this ratio varies by basin, a ratio less than one is generally indicative of oil and gas emissions (COGCC, 2007; Gilman et al., 2013; Prenni et al., 2016; Swarthout et al., 2013, 2015). During FRAPPÉ, Halliday et al. (2016) reported an i- to n-pentane ratio of 0.89 at PAO. This ratio is expected to be enhanced in areas where fuel evaporation is an important source of NMHCs (Harley et al., 1992; McGaughey et al., 2004), as i-pentane is an abundant component of gasoline and is elevated relative to n-pentane. As such, ratios in urban areas are often in the range of 1.5-4 (Baker et al., 2008; McLaren et al., 1996; Parrish et al., 1998; Russo et al., 2010a). For ROMO-LP hourly samples, shown in Figure 7, the average ratio of i-pentane to n-pentane was 1.10 throughout the campaign. The transect samples had an average ratio of 1.19 (not shown), which is 8% higher than the ROMO hourly samples. The slightly higher values along the transects may reflect impacts from nearby vehicle emissions. In Figure 7, markers are coloured by ethane mixing ratio. As seen in Figure 7, a strong oil and gas signature is evident on many sample days, with low i- to n-pentane ratios associated with higher ethane mixing ratios; similarly, higher ratios were usually associated with lower ethane mixing ratios from clean air masses transported from the west.

Figure 8 shows the spatial and temporal variability of i- to n-pentane in all of the canister transect samples, with the points sized by ethane mixing ratio (other gases are shown in the supplement Figure S3). The transect samples show a general east-west gradient with lower ratios typically observed at the easternmost sites, consistent with the oil and gas production activity in this area. However, significant variability was observed throughout the study at most of the transect sampling locations, with the range encompassing conditions indicative of dominant oil and gas influence (<1), to conditions more indicative of urban influences (1.93). On three occasions samples were collected from the Front Range up to the ROMO-TR site near the continental divide. Even limiting measurements to these three sampling dates at high elevation, the i- to n-pentane ratios indicated a range of air mass source histories impacted the park.

## 4. Ozone

### 4.1 Ozone Distributions at ROMO during FRAPPÉ

Ozone measurements at the ROMO-LP and ROMO-TR sites show coincident peaks for most of the FRAPPÉ study period (Figure 3), although $O_3$ mixing ratios were slightly higher and had considerably less variability at the ROMO-TR site. For ease of comparison, the frequency distributions of the $O_3$ mixing ratios are shown in Figure 9. The mean $O_3$ mixing ratios at ROMO-TR and ROMO-LP were 53 ppb (1-hr max: 82 ppb) and 43 ppb (1-hr max: 79 ppb), respectively, where ROMO-LP is lower in elevation and closer to the Front Range. Differences in these distributions can be attributed to several factors. First, the Longs Peak site is located in a valley, where pollutants can be trapped by stable nocturnal inversions, allowing efficient dry deposition or the titration of $O_3$ by NO, as seen by the regular night time decreases in $O_3$ in Figure 3. In comparison, the Trail Ridge Road site near the Continental Divide sits above the nocturnal inversion and is isolated from surface loss processes. This results in a skewed distribution toward lower mixing ratios at the Longs Peak site, with the median, 25th and 75th percentile values of 45 ppb, 36 ppb and 51 ppb, compared to the normal distribution at Trail Ridge Road with higher median, 25th and 75th percentile values of 53 ppb, 49 ppb and 57 ppb (Figure 9). Second, additional photochemical processing can occur during transport to the more distant Trail Ridge site during upslope events, resulting in slightly higher $O_3$ levels, as indicated by the 1-hr maximum values (ROMO-TR = 82 ppb; ROMO-LP = 79 ppb). Nevertheless, the relationship of the timelines of the $O_3$ data from both sites during elevated concentrations suggests they are often impacted by similar air masses ($r^2$=0.33, $p$<0.0001).

Elevated $O_3$ mixing ratios at ROMO are generally expected to be related to air masses that are transported to the site from the NRFMA resulting from the mountain-plains solenoid circulation, a local transport phenomenon occurring frequently during summertime (Bossert and Cotton, 1994; Sullivan et al., 2016; Toth and Johnson, 1985). Such conditions have the potential to transport air pollutants into the park from oil and gas source regions as well as from urban sources across the Front Range, as shown in Figures 4 and 5. Conducting a similar back trajectory analysis for $O_3$ as done for the VOCs, Figure 10 shows results for HRT-ORT for the top 10% of $O_3$ measurements. As for the VOCs, elevated $O_3$ mixing ratios are largely associated with transport from the NFRMA (including Denver).

**4.2 High O$_3$ Days**

Impacts to ROMO during upslope events can be quite pronounced with elevated mixing ratios, but assessing these same upslope events averaged over a day or more diminishes their visibility resulting in lower overall values for these longer time periods. Therefore, we use an event based approach focused on upslope events from the NFRMA to examine what

sources of Front Range emissions are impacting ROMO. During FRAPPÉ, 1-hr O$_3$ mixing ratios equal to or greater than 70 ppb were observed at both the ROMO-LP and ROMO-TR sites. These events are indicated by the grey shaded areas in Figure 3 and are summarized in Table 2. Elevated O$_3$ mixing ratios were typically accompanied by increased concentrations of various VOC source tracers. For example, alkyl nitrate concentrations (2-butyl nitrate, 2-BuONO$_2$, is shown as an example) in Figure 3 were elevated on six of the seven high O$_3$ days (not August 23). Alkyl nitrates are produced from the oxidation of

their parent hydrocarbons such as n-butane and n-pentane, emissions derived primarily from fossil fuels in the NFRMA, and share a common photochemical production mechanism with O$_3$. PAN was also elevated (>700 pptv) on all of the days shown in Table 2 except 23 and 24 August (see below).

During upslope flows created by thermally driven mountain-valley circulation, a mixture of direct emissions and photochemically produced secondary compounds can be transported to ROMO. We observed increases in concentration of a

mixture of photochemically produced (PAN, alkyl nitrates) and directly emitted (alkanes, NO$_x$) compounds at ROMO during these events. The photochemistry that produces species like PAN and alkyl nitrates can occur anywhere along the transport path to the ROMO receptor site. VOC signatures are particularly useful in assessing the influence of oil and gas emissions in the Front Range on air masses with elevated O$_3$ mixing ratios transported to the park. As noted above, the i- to n-pentane ratio is expected to be less than one for air masses primarily impacted by oil and gas emissions. Two O$_3$ episodes, August 8 and

August 18, occurred when the i- to n-pentane ratios were less than one at ROMO-LP (Table 2) together with elevated levels of alkyl nitrates. As seen in Figure 3, these dates also had some of the highest ethane mixing ratios observed throughout the study period, in addition to elevated mixing ratios of halocarbons, OVOCs, and ethyne. Although the air mass VOC signature does reflect several Front Range source types, it is dominated by oil and gas emissions as evidenced in the i-pentane to n-pentane ratio.

To further investigate the impact of VOCs on $O_3$ we next examine photochemical products and tracers. PAN is an excellent tracer of photochemical activity because it has a low background abundance and it is not directly emitted. Zaragoza et al. (2017) showed that when PAN and $O_3$ are disconnected in the Front Range, the source of the $O_3$ is not local photochemical production. At ROMO we see that increases in $O_3$ are associated with increases in alkyl nitrates and PAN suggesting a common

photochemical source. Zaragoza et al. (2017) also showed that regional mixing is common during the most photochemically active summertime days in the Front Range. Emissions from multiple sources in the region are injected into air masses as they circulate in the Front Range and secondary species reflect production driven by a mix of sources. Peak PAN mixing ratios at BAO and ROMO were very similar on both August 8 and 18 (~800 pptv), supporting the assumption that mixed Front Range air pushed up into ROMO on these particular days. PAN/PPN (propionyl peroxynitrate) ratios at BAO during the afternoon

on 18 August (high $O_3$ day at ROMO) ranged from 0.17 - 0.26. These relatively high ratios are indicative of a large PAN and PPN source from the oxidation of alkanes from oil and gas production at BAO and throughout the region (Lindaas et al., submitted; Zaragoza et al., 2017).

Analogous to PAN, alkyl nitrates are photochemically produced simultaneously with ozone in the atmosphere. Here, we use coincident observations of alkyl nitrates during these elevated $O_3$ periods to estimate the contribution of the light

alkanes from oil and gas emissions to $O_3$ production. Alkyl nitrate formation can be used as a proxy for $O_3$ production if we make the following assumptions. First, their photochemical production is rapid compared to removal processes and mixing across gradients. The rate of alkyl nitrate formation is tied to its parent alkane's concentration and reaction rate with the hydroxyl radical (OH), which is the hydroxyl radical reactivity (R1), as this is the rate limiting step in alkyl peroxy radical ($RO_2$) formation. In NFRMA areas that are influenced by oil and gas emissions, the hydroxyl radical reactivity values for the

alkyl nitrate parent alkanes are high (Figure 11), resulting in rapid production compared to their removal rates. Because of their relatively long lifetimes (~ ≤10 days for summer), the $C_2$-$C_5$ alkyl nitrates can be transported long distances and serve as a temporary reservoir for $NO_x$, ultimately leading to $O_3$ production in downwind or remote regions (e.g., Clemitshaw et al., 1997; Flocke et al., 1998; Roberts et al., 1998). For the upslope events on August 8 and 18, the assumption that photochemical production of the alkyl nitrates is rapid compared to removal processes is supported based on the rates of formation (hours)

versus their removal (days). Second, the precursor compounds (VOCs and $NO_x$) both have sources in Weld County, where

significant oil and gas development exists $NO_x$ was elevated during both events (not shown) and the 2014 National Emissions Inventory (US EPA, 2017) suggests Weld County has the largest emissions of $NO_x$ and VOCs in the state, and so it is reasonable to assume that emissions are collocated. The collocation of sources indicates these air masses likely contained sufficient $NO_x$ for alkyl nitrate and $O_3$ formation and the excess $O_3$ is clearly related to VOC and $NO_x$ emissions from the oil

and gas region. There could be $NO_x$ additions to the air mass as it moves west, but this does not violate the assumption because of the timescale for the transport, which is relatively fast. Third, transport time from the source region (NFRMA) to the park is rapid (on the order of hours). This assumption is validated by the time series plots shown in Figure 13, where during the upslope events, rapid increases in mixing ratios are observed within an hour of when the wind shifts to an upslope direction.

In general, this approach is similar to estimating $O_3$ production efficiency and background $O_3$ using $O_x/NO_z$ but

instead we use the individual alkyl nitrates in $NO_z$ because of the abundance of alkanes emitted from oil and gas operations. Moreover, we can better apportion the photochemical processing from the source region emissions because we are using an ensemble of individual compounds that have known rate constants, branching ratios and processing times, as opposed to using a bulk parameter such as a $NO_z$, where the composition is not accurately known. For the oil and gas influenced upslope events at ROMO, PAN, $O_3$, $NO_x$, and alkyl nitrates all show coincident increases (Figure 3; Benedict et al., 2018). For the August 8

and 18 events, the overall air mass composition (Figures 3, 12, 1S and 2S) and photochemical age (Figure 13) indicate that the source region emissions and processing times in the Front Range were comparable and these assumptions are valid.

Because the production of $O_3$ and alkyl nitrates is tied to their common precursor, the alkyl peroxy radical ($RO_2$), correlation between $O_3$ and the $C_2$-$C_5$ alkyl nitrates is expected. As a first step, the y-intercept is used to estimate the background $O_3$ value for the event period prior to the start of the upslope event. As shown in Table 3, each of the individual

alkyl nitrates gives a slightly different background $O_3$ value (y-intercept); however, they are all within a few ppbv of each other, providing a reasonable estimate of the background $O_3$ during that time period. The values reported for this work are also in the range of values reported by McDuffie et al. (2016) for BAO during summer 2014. The main source of the $C_2$-$C_5$ alkyl nitrates is the photochemical production from their parent alkane precursors, as outlined in the following simplified reaction scheme:

$$RH + OH \rightarrow R + H_2O \qquad\qquad (R1)$$

$$R + O_2 \rightarrow RO_2 \qquad\qquad (R2)$$

$$RO_2 + NO \rightarrow RO + NO_2 \qquad\qquad (R3a)$$

$$RO_2 + NO + M \rightarrow RONO_2 + M \qquad\qquad (R3b)$$

$$NO_2 + h\nu \rightarrow NO + O \qquad\qquad (R4)$$

$$O + O_2 + M \rightarrow O_3 + M \qquad\qquad (R5)$$

As shown by Reactions 1-5, alkyl nitrates share a common photochemical production mechanism with $O_3$. However, $O_3$ formation results from the photolysis of $NO_2$ (R4) whereas the formation of alkyl nitrates serve as a sink for $NO_x$, RO and $RO_2$, which affects $O_3$ production efficiency (Atkinson et al., 1982; Ranschaert et al., 2000; Russo et al., 2010b). Nonetheless, based on these reactions, it is expected that under a wide range of conditions $O_3$ and alkyl nitrates should be correlated as they are produced concurrently in the atmosphere (e.g. Abeleira et al., 2018; Day et al., 2003; Flocke et al., 1991; Perring et al., 2013; Rosen et al., 2004; Russo et al., 2010b). Methyl nitrate is excluded from this analysis because photochemical production from its parent alkane, methane, does not meet our assumptions, and there are primary sources which impact the park (Benedict et al., 2018). For both of the oil and gas influenced upslope events, $O_3$ is plotted against each of the alkyl nitrates for the time periods immediately before and after the event, as shown in Figure 14 for 2-butyl nitrate. This relationship can be used to provide a rough estimate of the excess $O_3$ associated with the upslope events transporting oil and gas impacted air masses from the NFRMA to the park. Using this method we can also investigate the contribution of other VOCs to $O_3$ production; however, in the case where alkanes are the dominant contributor to OH reactivity, we focus on the alkyl nitrates produced photochemically to provided an estimate of oil and gas impacts on $O_3$. As noted in Rosen et al. (2004), only a limited number of studies report the correlation of $O_3$ or $O_x$ with alkyl nitrates, with half of the studies using the sum of alkyl nitrates ($\Sigma$ANs) because a non-selective technique was used that did not allow for individual speciation of the alkyl nitrates (Flocke et al., 1991; Neuman et al., 2012; O'Brien et al., 1995; Perring et al., 2013; Roberts et al., 1996; Rosen et al., 2004). Flocke et al. (1991) reported the correlation of $\Sigma C_2$–$C_5$ straight chain alkane-derived nitrates plus a single $C_4$ branched chain alkane-derived nitrate with $O_x$ measured at Shauinsland, Germany.

For both events (August 8 and 18), when the alkanes and OVOCs both contributed an average of 40% to OH reactivity, the average background $O_3$ was 43±2 ppbv. We use the background $O_3$ value to estimate the range of influence on $O_3$

production derived from oil and gas emissions driven by the individual parent alkane precursors (attributed $O_3$). To this end, the orthogonal distance linear regressions between $O_3$ and each of the individual $C_2$-$C_5$ alkyl nitrates are used. To estimate the excess $O_3$ produced and transported to ROMO during the upslope events, the slope of $O_3$ versus each of the alkyl nitrates was determined, then the hourly concentration for each alkyl nitrate was multiplied by the slope value and the intercept value was subtracted. Table 3 provides a summary of the $O_3$/$RONO_2$ slopes in ppbv/ppbv, intercept values (background $O_3$) in ppbv, the range of attributed $O_3$ (ppbv), and the attributed $O_3$ value (ppbv) based on the correlation in each of the $O_3$-alkyl nitrate scatter plots. The estimated average attributed $O_3$ production values determined for each event are 22±3 ppbv and 19±2 ppbv for the August 8 and August 18 events at ROMO-LP, respectively. These values represent an upper limit on the excess $O_3$ produced from NFRMA air masses transported to ROMO that were dominated by oil and gas emissions. Other compounds produced photochemically in situ (e.g., PAN and acetone) provide similar values to the alkyl nitrates for the attributed $O_3$ (not shown). The alkyl nitrate correlation method does not allow us to explicitly separate the impact of $NO_x$, additional OVOCs or other species such as alkenes (Figure 11), which all likely contribute to ozone production. These results are consistent with measurements along the NFRMA during FRAPPÉ by Cheadle et al. (2017), who showed that on individual days emissions from oil and gas operations can contribute upwards of 30 ppbv to excess $O_3$ production. The high concentration air masses measured in the Front Range contain a similar mix of compounds as the air masses transported to ROMO during upslope events. Accounting for mixing and dilution, enhancements in $O_3$ resulting from oil and gas emissions are expected to be lower at ROMO. Worth noting is the consistency of these $O_3$ enhancements for the two events at ROMO and the marked similarity of the air mass compositions measured in the park (Figure 12). Figure 12 shows $O_3$, ethane and alkyl nitrate mixing ratios during the events, along with meteorology data. For the August 8 upslope event, the winds were variable and out of the SSE and it was sunnier (solar radiation ~200-300 W $m^{-2}$ higher) than August 18, but ~3 °C cooler (not shown). The August 18 upslope event had winds from the east and a sharp drop in solar radiation just prior to the upslope event. Nonetheless, the alkyl nitrates, light alkanes, PAN, and $O_3$ mixing ratios were strikingly comparable during these two events (Figure 12) as were the overall air mass reactivity (Figure 11) and photochemical age (Figure 13), indicating comparable source region emissions. When considering the entire study period, and not just these specific events, the oil and gas contribution to $O_3$ at

ROMO is much smaller than during these episodes, given that upslope (south easterly) flow is not the predominant wind direction (McDuffie et al., 2016).

Intermediate i- to n-pentane ratios (1.12-1.44) were observed for measurements on July 22, August 11, and August 12, suggesting that multiple Front Range sources impacted ROMO during these $O_3$ events.  In addition to urban sources, biomass burning emissions influenced the Front Range on August 11-12 (Dingle et al., 2016).  The highest $O_3$ levels across the Front Range during FRAPPÉ were observed on July 22 (Pfister et al., 2017; Sullivan et al., 2016), which also had the highest average afternoon (12:00-20:00) concentrations of total alkanes (9700 pptv), alkenes (277 pptv), biogenics (252 pptv), alkyl nitrates (110 pptv), and OVOCs (9.8 ppbv; methanol, acetaldehyde, MVK+MACR, MEK, and acetic acid) at ROMO among all of the high $O_3$ days.  Throughout the region, clear skies and a lower and slow growing planetary boundary layer, coupled with reduced ventilation, resulted in less dilution and increased precursor concentrations, contributing to the elevated $O_3$ (Pfister et al., 2017).  These elevated mixing ratios were likely then transported to ROMO by thermally driven upslope flow observed along the Front Range (Sullivan et al., 2016).

The $O_3$ events observed on August 23 (ROMO-LP) and August 24 (ROMO-TR) have neither an oil and gas nor an urban signature. PAN also remained < 300 pptv during both days. Here we focus on the event on August 23 at ROMO-LP, where VOC measurements were collocated.  On August 23, air masses came from the west without passing over the NFRMA and had much lower abundances of $O_3$ precursors and photochemical products (Figure 3), suggesting that the high $O_3$ mixing ratios were not generated from regional photochemical production.  These high $O_3$ mixing ratios are associated with the red region in western Colorado in Figure 10.  Mixing ratios of alkanes (total 1505 pptv), alkenes (total 93 pptv), aromatics (42 pptv), alkyl nitrates (total 17 ppt), halocarbons (4 pptv; $C_2Cl_4$, $CH_3I$, $CH_2Br_2$, $CHBrCl_2$, $CHBr_3$), and OVOCs (2.8 ppbv; methanol, acetaldehyde, MVK+MACR, MEK, and acetic acid) were all low compared to the other high $O_3$ periods and most of the study.  For this same time period the average $NO_x$ mixing ratio (0.52 ppbv) was lower than those during other high $O_3$ periods (1.2-3.1 ppbv; no data for August 11).    Additionally, a comparison of soundings from Denver (http://weather.uwyo.edu/upperair/sounding.html) shows a pocket of dry air from 500 to 100 hPa which does not occur on any of the other high $O_3$ days (Figure 15a), and water vapour decreases during the event on August 23 (Figure 15b). This is the only high $O_3$ event for which $O_3$ and water vapour exhibit a robust anti-correlation ($R^2$=0.627, $p$<0.0001) during the study.

The low coincident mixing ratios of $NO_x$ and VOCs, the dry air aloft, and the robust anti-correlation of $O_3$ and water vapour at the surface during the $O_3$ event suggest an intrusion of stratospheric or upper tropospheric air on August 23. A stratospheric-tropospheric exchange event also was observed in Fort Collins in early August during FRAPPÉ (Sullivan et al., 2016).

### 4.3 Hydroxyl Radical (OH) Reactivity and Photochemical Age

Photolysis of $NO_2$ is the only way to produce $O_3$ in the troposphere. Oxidation of VOCs by hydroxyl radicals (OH) in the atmosphere is typically the first step in the formation of peroxy radicals ($RO_2$) which react with NO to produce $NO_2$. Calculating the kinetic OH reactivity (OHR) helps to identifiy compounds that are likely to quickly form $RO_2$ and ultimately contribute to $O_3$ production. The OHR for VOCs is calculated using the following expression:

$$OHR = \sum [VOC]_i \times k_{VOC_i+OH} \quad \text{Eq. (1)}$$

where $[VOC]_i$ is the concentration of each VOC and $k_{VOC_i+OH}$ is the reaction rate constant for each VOC with OH. The OHR is calculated for the high $O_3$ events during FRAPPÉ which had an oil and gas influence, a combined oil and gas and urban influence, the stratospheric/upper troposphere intrusion event, and for background conditions (without regard to the $O_3$ mixing ratio and defined as lowest 10th percentile for ethane). This metric is an imperfect measure of $O_3$ production potential since it does not account for chain termination or propagation steps, nor does it properly capture differences in VOC production of

peroxy radicals during photolysis or reaction with other oxidants. However, OHR is a useful tool because it is a measure of the initial rate of peroxy radical formation that can be interpreted as the potential for a specific compound to ultimately produce $O_3$.

For the events influenced by oil and gas emissions, the alkanes and OVOCs were the major contributors to the VOC-OHR, comprising almost 40% and 39%, respectively, of the calculated total (Figure 11). For the high $O_3$ air masses that also

included an urban influence, the OHR for OVOCs dominated (40%), followed by alkanes (~20%) and alkenes (~17%) which were more comparable, reflecting increased reactivity contributions of the light alkenes from urban areas. However, the alkanes were still the largest non-OVOC contributor of OH reactivity to this category. The importance of isoprene, with ~21% of the total reactivity, is also more apparent, even though the average mixing ratio was only 64 pptv for these days. Larger mixing ratios of isoprene (~270 pptv) during background periods results in the calculated VOC-OHR being dominated (~74%)

by isoprene reactivity and of similar magnitutude as the more polluted oil and gas days. In all cases, ethyne and the aromatics

were only minor contributors to the calculated overall OHR. Finally, it is worth noting that the OHR values calculated here are lower bounds on the actual VOC-OHR in the studied air masses. The OHR values only represent the initial attack of OH with the VOC and does not reflect the subsequent contributions of the oxidation products formed and how these influence and change the overall air mass and its reactivity.

5         To estimate the impact of the oil and gas emissions on OHR at the park, we compare the average straight chain alkane OHR during the upslope events to that of the background and combined oil and gas and urban-influenced alkane OHRs, which include the $C_2$-$C_8$ alkanes. The reactivity attributed to the alkanes is considered because they are the dominant VOCs associated with the oil and gas emissions. The alkane OHR for the oil and gas events was 0.62 $s^{-1}$, which comprised ~40% of the total calculated OHR (includes all measured compounds) for those events. In comparison to the oil and gas influenced events, the

alkane OHR for the urban-influenced events was only 0.21 $s^{-1}$, while the background alkane OHR was only 0.02 $s^{-1}$, and comprised much smaller fractions of the total calculated OHR for those periods. The difference in alkane reactiviy between the oil and gas events and the urban-influenced events was 0.41 $s^{-1}$; the difference between oil and gas influenced and background events was 0.60 $s^{-1}$. Furthermore, the percent contribution of the individual straight chain $C_2$-$C_8$ alkanes to the total alkane OHR is similar to that of other measurements made in the regions across seasons (Abeleira et al., 2017; Halliday

et al., 2016; McDuffie et al., 2016; Swarthout et al., 2013) underscoring the consistent temporal and spatial compositional influence of the oil and gas source signature impacting ROMO during upslope events (Table 4).

        Air mass photochemical age can be calculated using the alkyl nitrate to parent hydrocarbon ratio (R-ONO$_2$/R-H) (e.g. Bertman et al., 1995; Russo et al., 2010b; Simpson et al., 2003; Swarthout et al., 2013). This reaction scheme assumes only OH-initiated photochemical production of a given alkyl nitrate from its parent hydrocarbon, and so deviations from the model

suggest additional sources or chemistry not captured in this simple scheme. Details of the chemical mechanisms and calculations can be found in the supplement. The modelled ratios of 3-pentyl nitrate to n-pentane versus 2-butyl nitrate to n-butane are calculated using [OH] = $10^6$ molecules $cm^{-3}$ and are shown as the solid line in Figure 13. Measured ratios of these species are shown as the symbols and are coloured by $O_3$ mixing ratio. Based on this model, VOC ages in sampled air masses ranged from several hours to two days and, as generally observed with the pentyl nitrates, they typically lie below the model

prediction line with a relatively small offset. Focusing on the high $O_3$ periods, shown as red filled symbols, the oil and gas

influenced events on August 8 and August 18 show the best agreement with the model, with a photochemical age of less than one day, indicating local sources of the parent hydrocarbons and that photochemical production was driving the $O_3$ and alkyl nitrate production these days. On July 22, August 11 and August 12, when both urban and oil and gas impacts were observed, the measured ratios deviate from the model in a systematic way, suggesting that the air masses were influenced by multiple

chemical processes. Finally, on August 23 and 24, the alkyl nitrate to parent alkane ratios deviated from the photochemical line significantly, providing another line of evidence that the increase in $O_3$ was not from photochemical production in a polluted environment, consistent with influences of $O_3$ enriched air from the upper troposphere.

**5. Summary and Conclusions**

Rocky Mountain National Park is a Class I area that is afforded the highest level of air quality protection.

Nevertheless, $O_3$ mixing ratios can reach levels that negatively impact vegetation and human health, particularly for older adults and people who are active outdoors. In this study, $O_3$ and a suite of VOC measurements were made at ROMO as part of the 2014 FRAPPÉ field campaign. Although average and peak $O_3$ mixing ratios were lower during FRAPPÉ compared to other recent summer seasons, one-hour mixing ratios exceeding 70 ppbv routinely measured in the park during this period. This study was aimed at better understanding sources of VOCs that impact the park and, particularly, what drives these high

ozone events.

ROMO is located just west of the NFRMA in Colorado, and the park can be impacted by a variety of emissions sources. Significant variability in the measured VOC abundances suggest that the sampling site was impacted by emissions from biogenic, urban, combustion, agricultural, and oil and gas sources during the study period. Light alkanes measured at ROMO, markers for oil and gas production, were significantly elevated relative to background levels, and at times reached

mixing ratios measured in some oil and gas source regions during upslope events (e.g., Swarthout et al, 2015). Mixing ratios reached 9800 ppt for ethane, 1158 ppt for i-butane, 2378 ppt for n-butane, 2095 ppt for i-pentane, and 1712 ppt for n-pentane.

Using VOCs as source markers, we determined that the elevated $O_3$ periods were primarily associated with upslope events. For the oil and gas impacted days, Front Range emissions contributed an upper bound of 20 ppbv excess $O_3$, having a major impact on total $O_3$ mixing ratios on those days. In air masses with increased urban influence, alkenes played a more important

role; however, even for these urban influenced air masses, a clear oil and gas signature was also present. During background periods, when air masses arrived from the west, $O_3$ was typically low and VOC -OH reactivity was dominated by isoprene (74%). Total VOC-OHR was of similar magntitude on the background (0.99 s$^{-1}$) and oil and gas influenced (1.26s$^{-1}$) days but alkanes dominated total VOC-OHR (40%) on oil and gas days. The mixed urban and oil and gas influenced days were

characterized by slightly lower total VOC-OHR (0.825 s$^{-1}$) and OH reactivity was dominated by OVOCs (40%), alkanes (20%) and alkenes (17%). In one distinct case, elevated $O_3$ was associated with air mass transport from the west. For this case, the abundances of VOCs and other secondary species were very low (Total VOC-OHR =0.16s$^{-1}$), and there was a clear anti-correlation of $O_3$ and surface water vapour, suggesting that an intrusion of stratospheric or $O_3$-rich upper-tropospheric air contributed to elevated $O_3$ measured at the surface.

Elevated $O_3$ in the Front Range and ROMO has been a persistent air quality issue. Findings from this study reaffirm that ROMO is impacted by a variety of anthropogenic emissions sources in the NFRMA. State and federal agencies have implemented regulations that have successfully reduced emissions from traditional sources (mobile, power plant, and other activities) gradually reducing $O_3$ since 2009 at a rate of -0.25 ± 0.13 ppbv yr$^{-1}$ in ROMO (McGlynn et al., 2018). Continued population growth and the competing effects of increasing emissions from oil and gas extraction appear to be diminishing the

impact of the regulations as emissions from emerging sectors or increased population are contributing to the high $O_3$ episodes. Management practices aimed at reducing high $O_3$ levels at the park thus cannot focus on controlling a single source, but instead must take a broader approach to focus on reductions in the traditional sources as well as these new sources. Recent work by Zhou et al. (in prep) suggest that emission reductions of both volatile organic compounds (VOCs) and nitrogen oxides (NO$_x$) could lead to effective $O_3$ mitigation in the Intermountain West. Additionally, identifying periods of stratospheric $O_3$ influence

should also be a priority when assessing which high $O_3$ episodes can be mitigated through emissions control.

**Data Availability**

The data used in this analysis are available in the FRAPPÉ/DISCOVER-AQ data archive (https://www-air.larc.nasa.gov/cgi-bin/ArcView/discover-aq.co-2014) in the Ground-Other category under the PIs Collett, Fischer, and Sive.

**Acknowledgements**

Support for this work was provided by the National Park Service. The assumptions, findings, conclusions, judgments, and views presented herein are those of the authors and should not be interpreted as necessarily representing the National Park Service. We recognize the contributions of the FRAPPÉ/Discover-AQ PIs (Gabriele Pfister-NCAR, Frank Flocke-NCAR, Jim Crawford, NASA) and the other FRAPPÉ/Discover-AQ funding sources (NSF, NCAR, Colorado Department of Public Health and Environment, NASA) for their support and contributions in organizing and directing the experiment including the flight and measurement planning, field operations, and maintenance of the data archive.

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

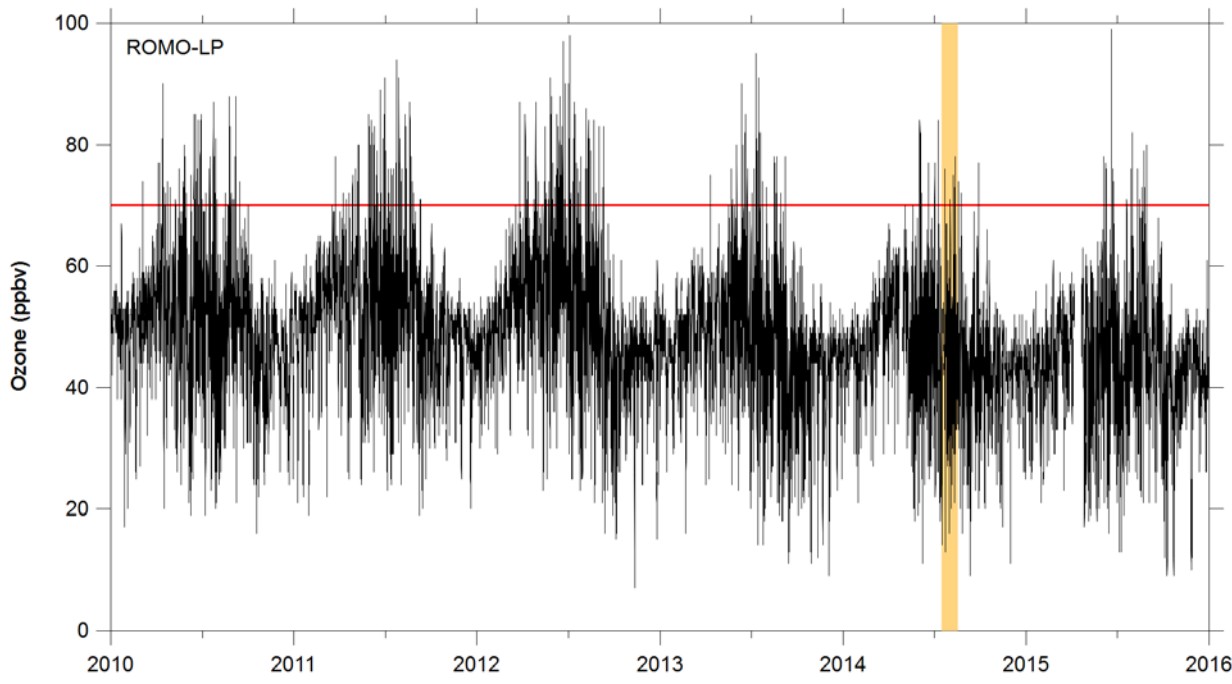

**Figure 1. Hourly O$_3$ measurements at ROMO-LP from January 2010 to December 2015. The red line indicates the current NAAQS 8-hr daily maximum value of 70 ppb. The shaded region designates the FRAPPÉ study period.**

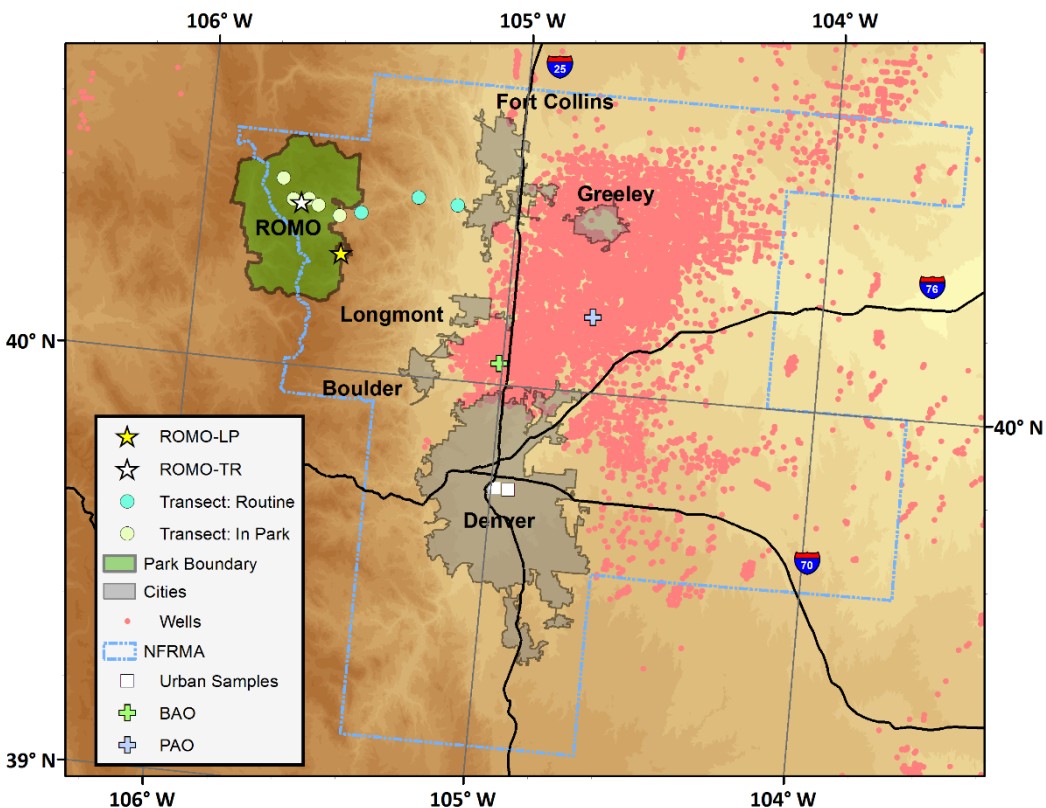

**Figure 2. Map of sampling sites, including locations of the transect samples. Also shown are oil and gas well locations (COGCC, 2018) and cities along the NFRMA.**

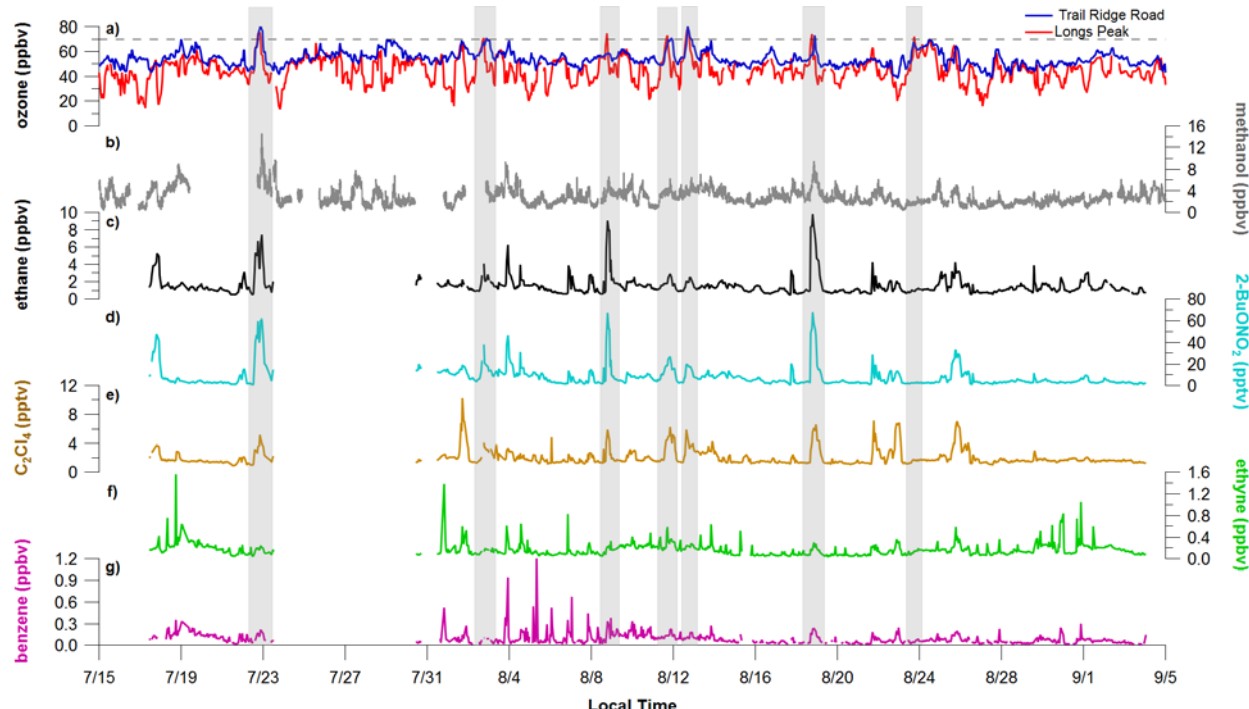

**Figure 3. A subset of VOCs measured at the ROMO-LP site from 17 July to 4 September 2014. From 19-22 and 23-30 July, the PTR-MS and GC system, respectively, were shut down due to air conditioner failures in the field laboratories. Ozone measurements from both the ROMO-LP (Longs Peak) and ROMO-TR (Trail Ridge Road) sites are also shown. Shaded areas represent periods selected for further analysis when hourly O$_3$ exceeded 70 ppb (dashed line).**

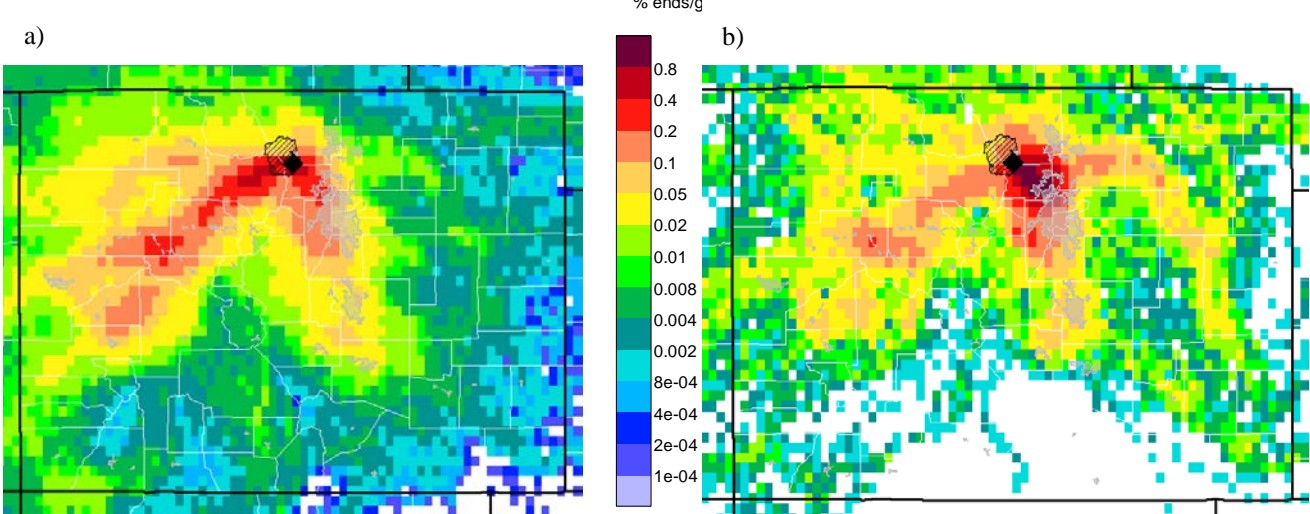

**Figure 4.** Residence time maps showing areas up to 2 days upwind of ROMO-LP (black diamond) during (a) all hours of the study period (July 17 – Sept 4, 2014) and (b) during hours with ethane > 90th percentile. Colours indicate the relative frequency of transport. Light grey overlay indicated urban areas. Units are the percent of total endpoints in each grid cell. Endpoints are upwind air mass locations calculated every hour for two days back in time for every trajectory that arrived at the receptor during the duration of the study period.

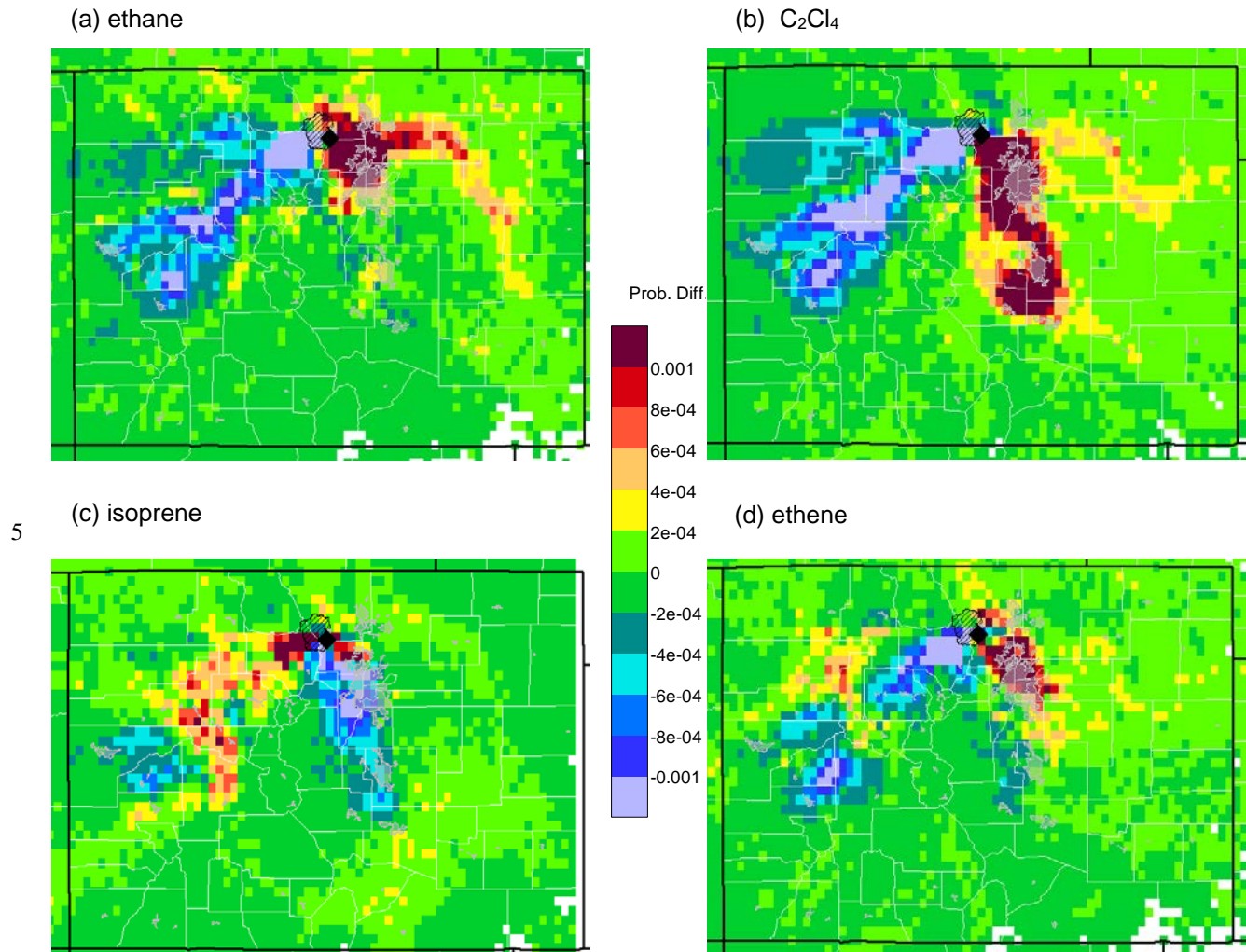

**Figure 5. Differences between upwind transport during hours of 90th percentile or higher mixing ratios and average conditions during the study (HRT-ORT) for (a) ethane, (b) C2Cl4, (c) isoprene, and (d) ethene. Yellow to red colours indicate areas that were more likely than average to be upwind during the 2 days prior to a high concentration measurement at ROMO-LP (black diamond). Blue to purple colours indicate areas that were less likely to be upwind, while areas in green had similar likelihoods of being upwind during average hours and during high mixing ratio hours. Light grey overlay indicated urban areas.**

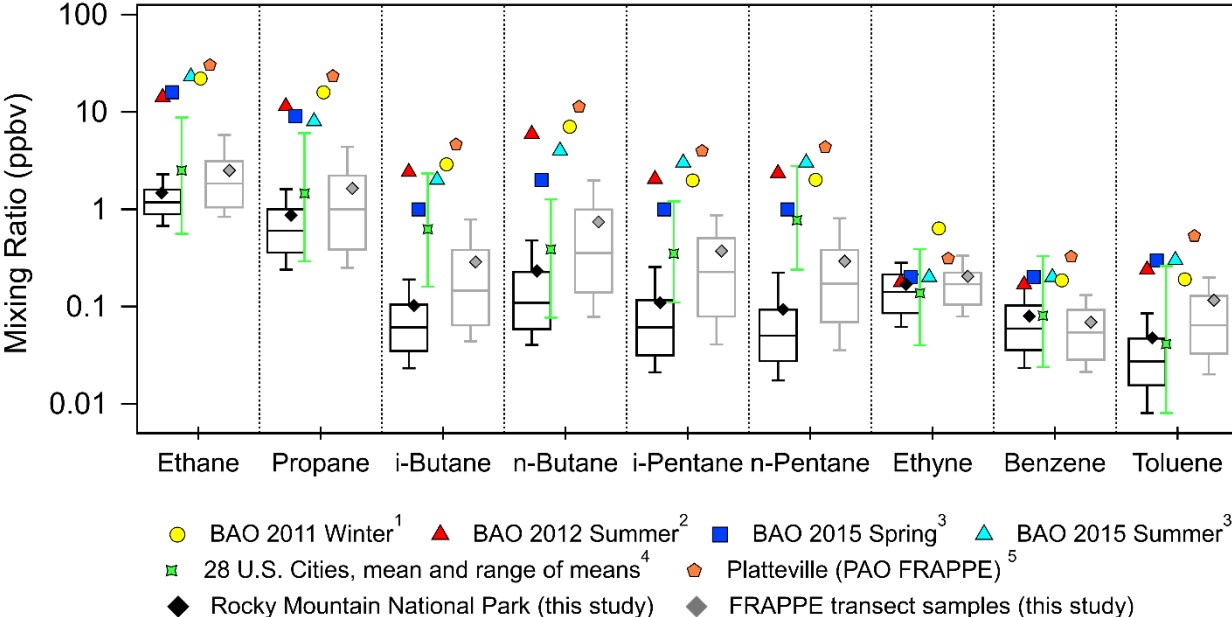

**Figure 6.** Box plots for mixing ratios of selected VOCs in ROMO hourly samples (black) and all transect canister samples (grey) and mean mixing ratios from previous studies ([1]Swarthout et al. (2013), [2]McDuffie et al. (2016) [3]Abeleria et al. (2017), [4] Baker et al. (2008), [5] Halliday et al. (2016)). Median mixing ratios from this study are shown as the black horizontal lines, the means are shown as the diamond symbols, the top and bottom of the boxes are the 75th and 25th percentiles and the whiskers are the 90th and 10th percentiles. The error bars on the Baker et al. (2008) data represent the maximum and minimum means observed across the cities. The data from Swarthout et al. (2013) are a mean of hourly canisters collected from noon on Feb 18, 2011- noon March 13, 2011. The McDuffie et al. (2016) data are a mean of in situ measurements made every 25 min from July 27 2012-August 12 2012. Abeleria et al. (2017) collected in situ hourly data and the values shown are means. PAO canisters were typically collected 2-3 times a day most days from 7/17-8/10, the data can be found in NASA Langley's data repository (https://www-air.larc.nasa.gov/data.htm).

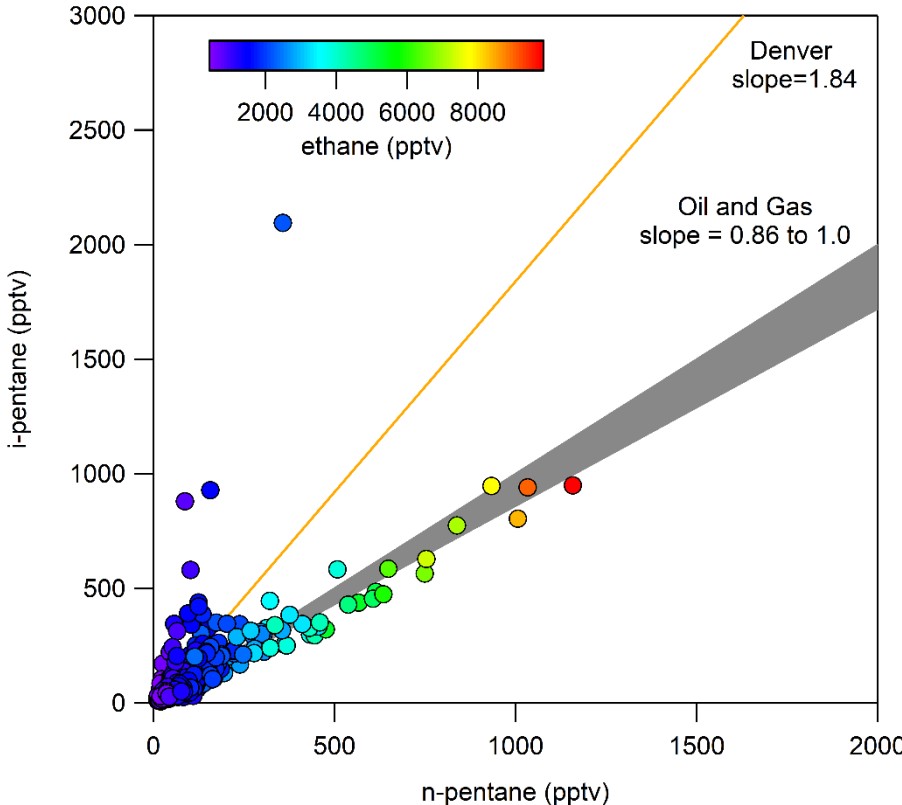

**Figure 7. Iso-pentane versus n-pentane for ROMO hourly samples, coloured by ethane mixing ratio. For reference, also shown are the i- to n-pentane ratio for measurements collected in Denver (Baker et al., 2008) and a range of slopes from 0.86 to 1 line where everything that falls in this range indicates oil and gas influenced (COGCC, 2007; Gilman et al., 2013).**

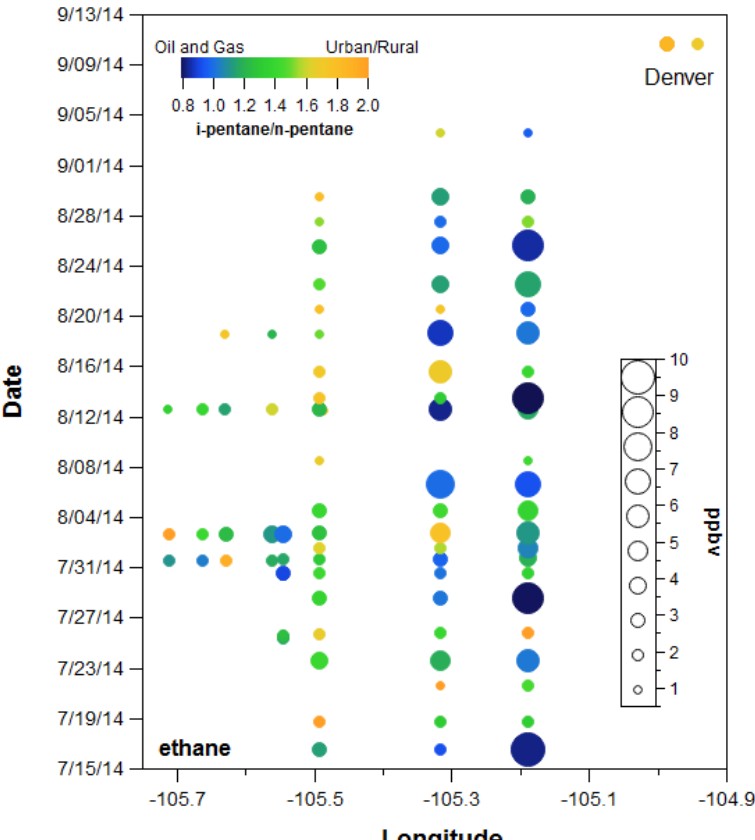

**Figure 8. VOC canister data plotted by longitude and date with ethane mixing ratio represented by the size of each point and the i- to n-pentane represented by the colour. Locations correspond to transect samples shown in Figure 2.**

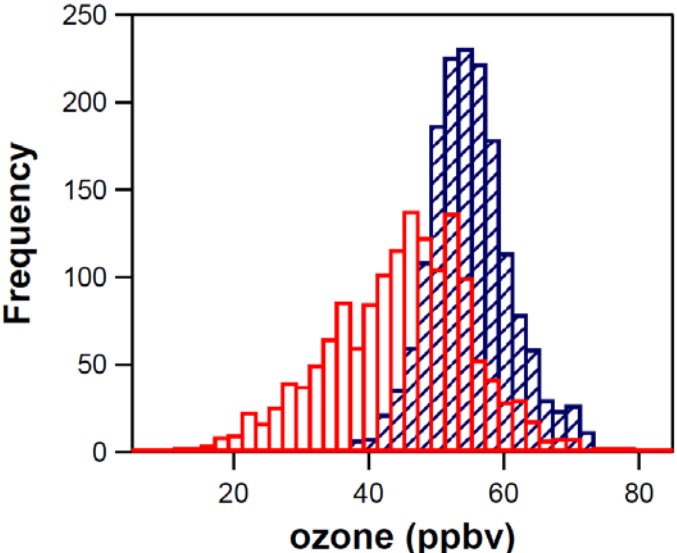

**Figure 9. Distribution of the O₃ mixing ratios at the ROMO-LP (red) and ROMO-TR (blue) sites for the entire study period.**

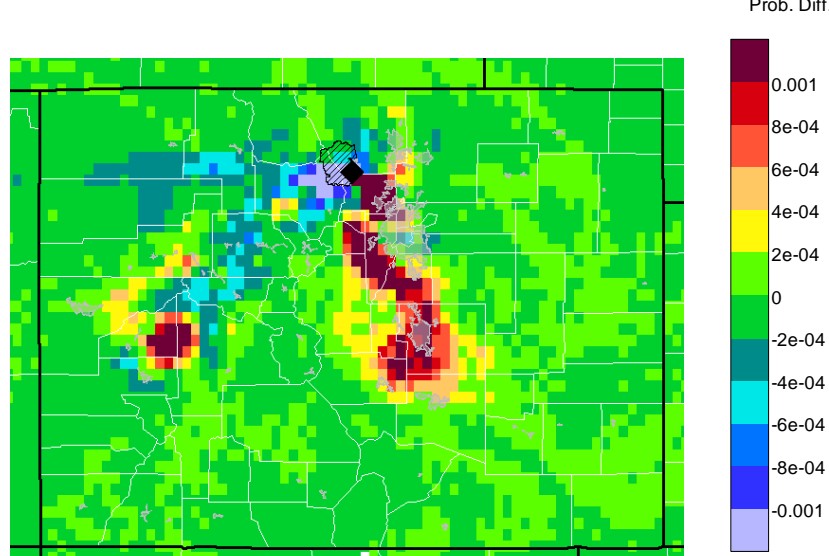

**Figure 10. Differences between upwind transport during hours of 90ᵗʰ percentile or higher mixing ratios and average conditions during the study (HRT-ORT) for O₃. Warm colours indicate areas that were more likely than average to be upwind during the 2 days prior to a high mixing ratio measurement at ROMO-LP. Cool colours indicate areas that were less likely to be upwind, while areas in green had similar likelihoods of being upwind during average and high mixing ratio hours.**

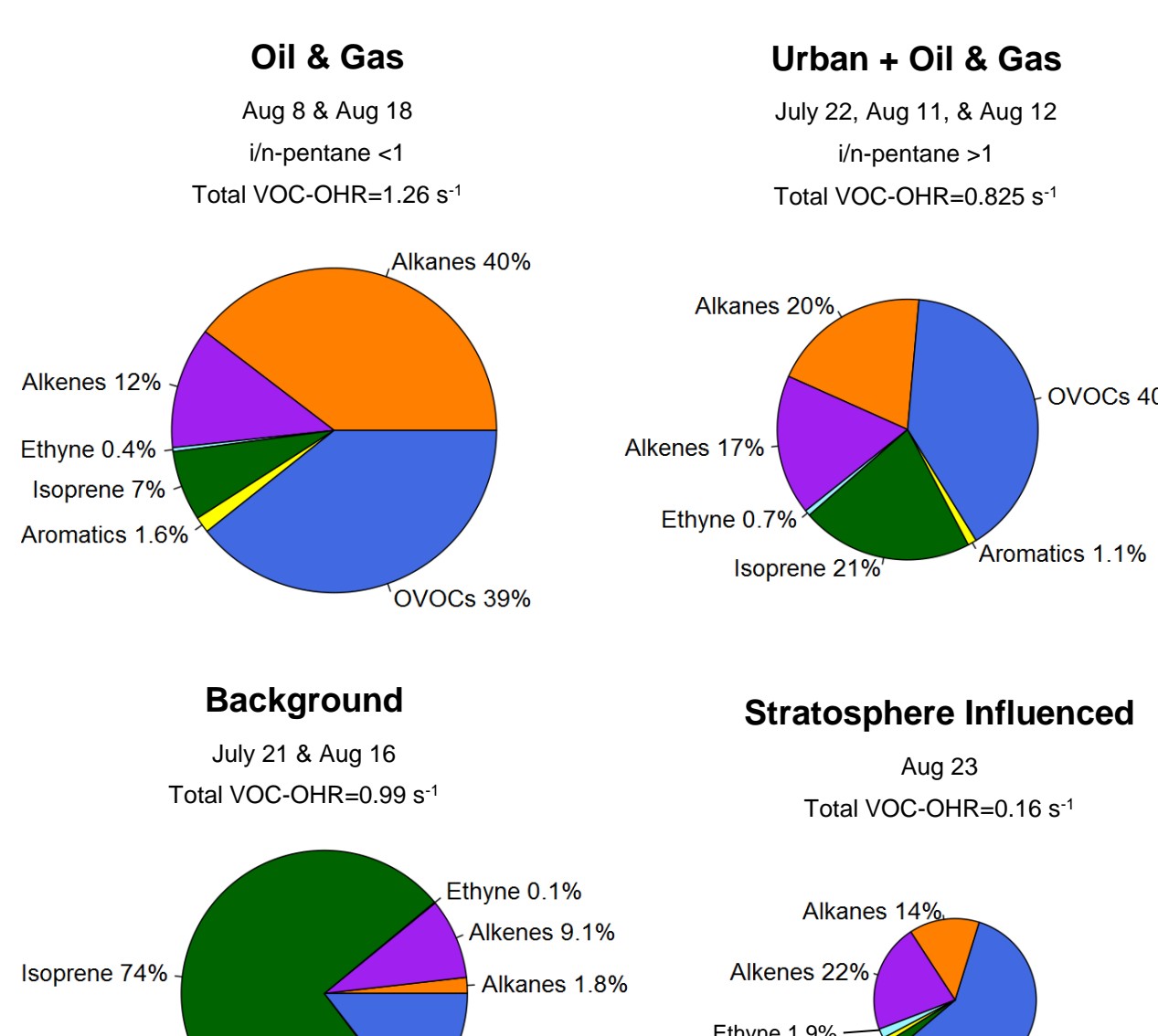

**Figure 11.** Contributions to •OH reactivity (OHR) for the high O₃ days for peak O₃ hours as shown in Table 2. Days with high O₃ were grouped by the ratio of i- to n-pentane for oil and gas influenced and urban influenced air masses. Background days, defined by the lowest 10th percentile for ethane, and the stratospheric influenced period are used for comparison to the high O₃ days resulting from anthropogenic activities. The size of each pie chart is proportional to the total OHR for the given type of air mass. Reactivity was calculated using rate constants from Atkinson and Avery (2003) and Atkinson (1986).

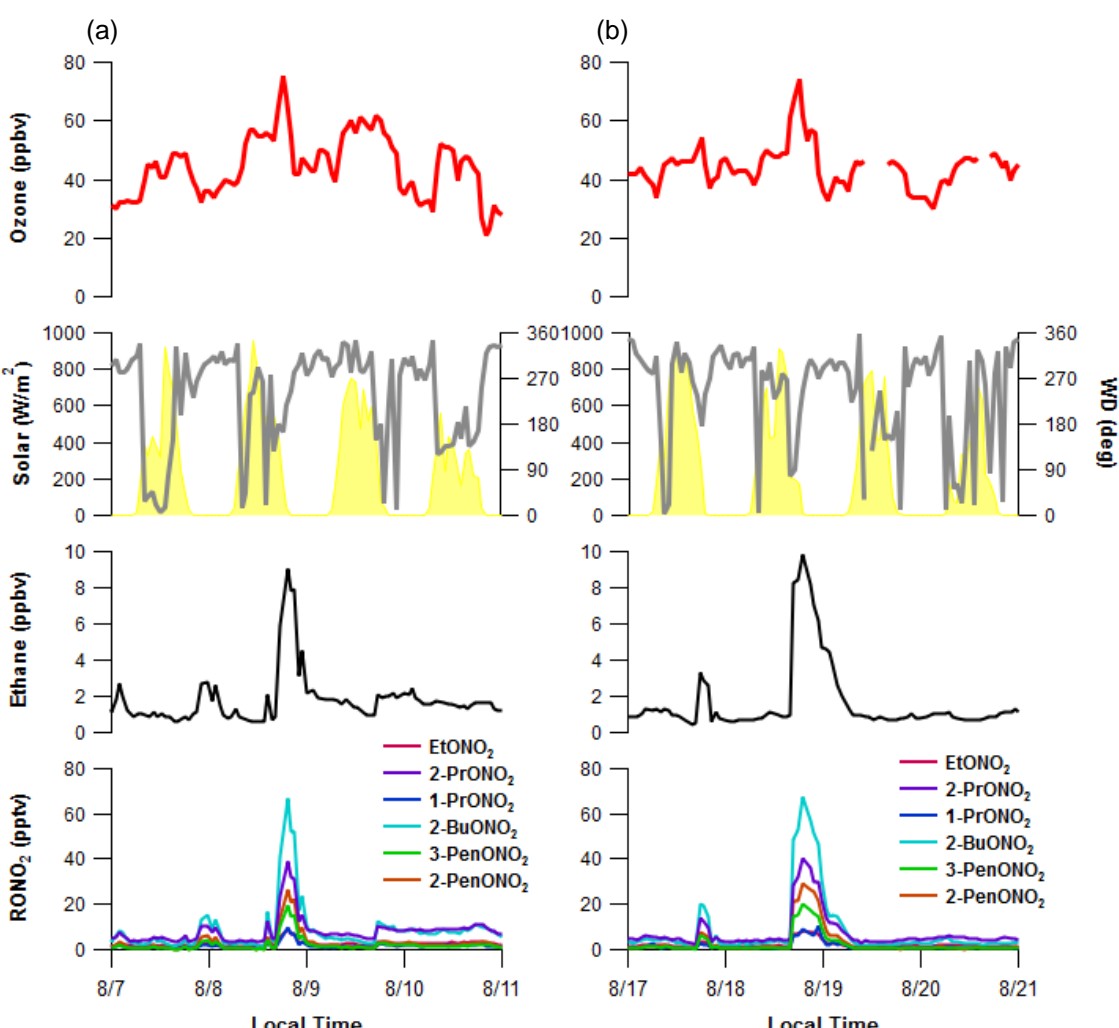

Figure 12.  Time series plots of $O_3$, solar radiation, wind direction, ethane, and the $C_2$-$C_5$ alkyl nitrates during the a) August 8 and b) August 18 upslope events.

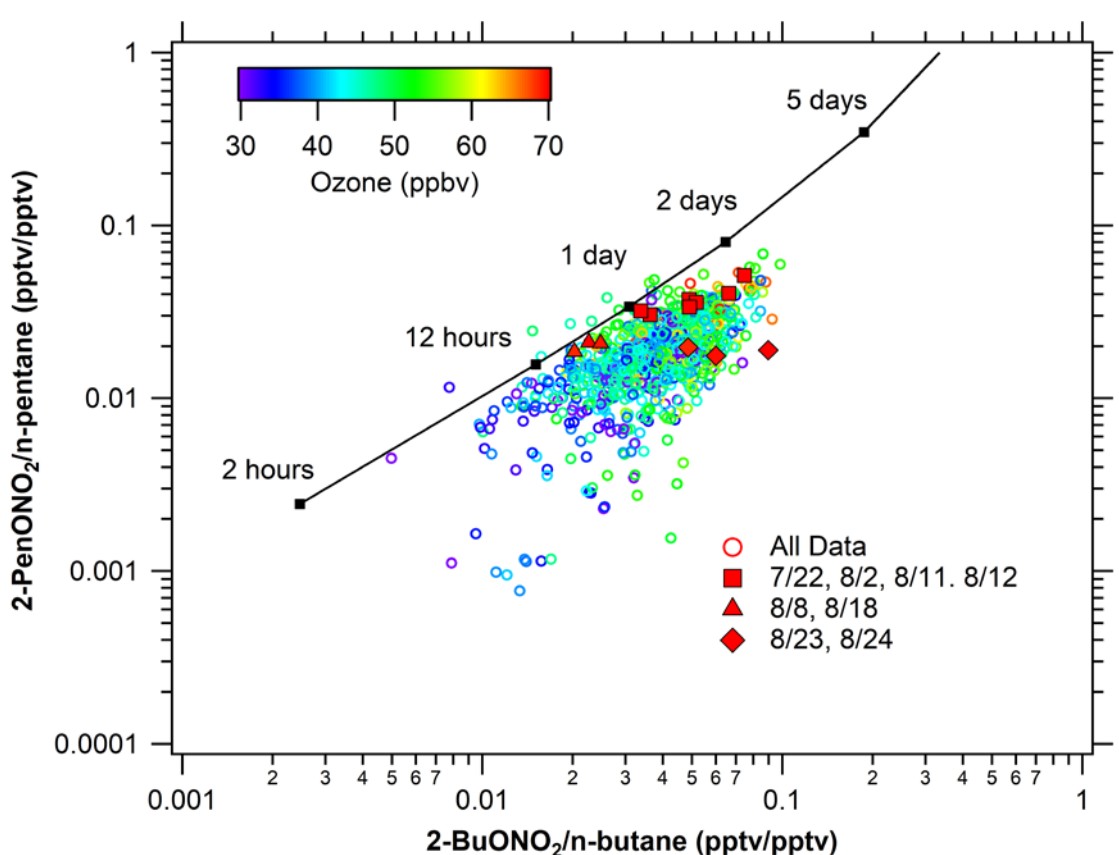

**Figure 13. Photochemical age of alkyl nitrates. Comparison of observed alkyl nitrate: parent hydrocarbon ratios to predicated values provide an estimate of photochemical age.**

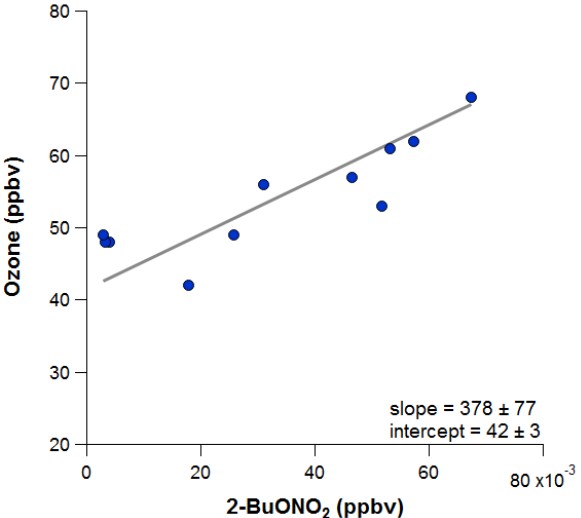

**Figure 14.** The orthogonal distance linear relationship of $O_3$ with 2-butyl nitrate during the August 18 event, when the park was clearly influenced by an air mass containing oil and gas emissions. This relationship is used in estimating the excess $O_3$ produced due to oil and gas emissions.

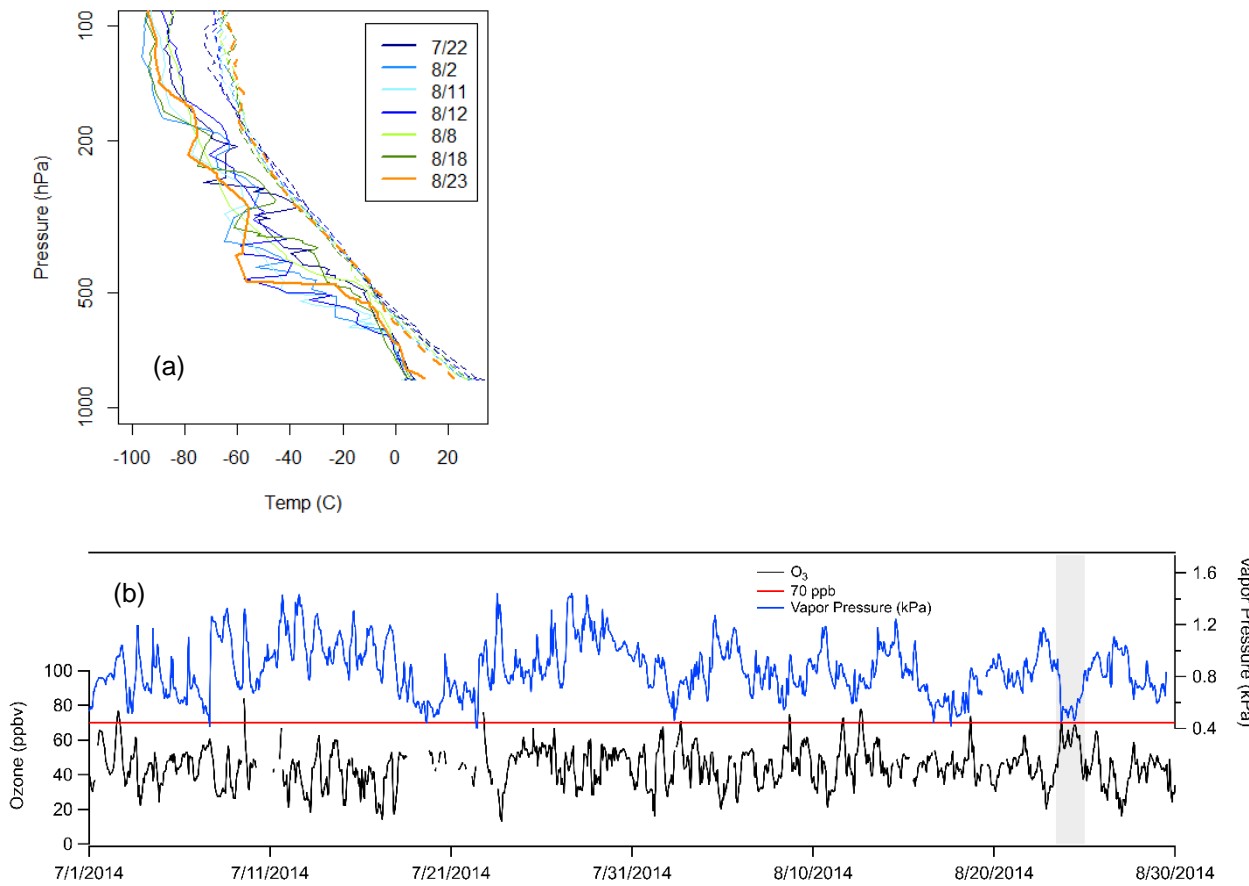

**Figure 15.  a) Denver soundings (http://weather.uwyo.edu/upperair/sounding.html) from high O₃ days, showing surface temperature (dashed lines) and dew point (solid lines).  b)  Measured O₃ concentration and water vapour at ROMO-LP during the study period. The event from August 23 is highlighted with grey shading.**

**Table 1. VOCs measured using the *in situ* GC system and PTR-MS at ROMO-LP. Also denoted are gases reported from the transect canister samples. Statistics are given for the GC and select PTR-MS data.**

| | | Average | Median | Max. | Min. | Standard Deviation | Measured in Canister Samples? |
|---|---|---|---|---|---|---|---|
| | | Mixing Ratio (ppt) | | | | | |
| ethane | GC | 1516 | 1255 | 9800 | 485.0 | 1061 | x |
| propane | GC | 872.0 | 616 | 6842 | 65.2 | 991 | x |
| i-butane | GC | 103.0 | 61.3 | 1158 | 11.1 | 141.0 | x |
| n-butane | GC | 233.0 | 110.0 | 2978 | 18.8 | 374.0 | x |
| ethyne | GC | 168.7 | 141.4 | 1547 | 17.3 | 129.4 | x |
| i-pentane | GC | 144.0 | 61.9 | 2095 | 7.3 | 168.0 | x |
| n-pentane | GC | 97.5 | 50.7 | 1712 | 8.7 | 151.0 | x |
| cyclopentane | GC | 7.5 | 4.9 | 148.0 | 0.6 | 9.7 | |
| n-hexane | GC | 34.0 | 16.0 | 1113 | 1.8 | 70.6 | x |
| n-heptane | GC | 15.6 | 8.7 | 396.0 | 0.5 | 29.4 | x |
| n-octane | GC | 13.4 | 6.4 | 304.0 | 1.0 | 26.4 | |
| n-nonane | GC | 10.6 | 5.7 | 151.0 | 1.1 | 17.7 | x |
| ethene | GC | 113.0 | 87.0 | 1497 | 26.3 | 111.0 | |
| propene | GC | 712.0 | 72.9 | 762.0 | 3.6 | 53.4 | |
| t-2-butene | GC | 4.7 | 3.2 | 80.6 | 1.0 | 5.7 | |
| 1-butene | GC | 6.5 | 5.2 | 84.8 | 1.1 | 6.0 | |
| i-butene | GC | 14.7 | 12.6 | 74.0 | 1.7 | 8.5 | |
| c-2-butene | GC | 5.6 | 3.4 | 76.3 | 1.0 | 7.1 | |
| isoprene | GC | 113.6 | 55.1 | 1559 | 1.0 | 185.5 | x |
| a-pinene | GC | 68.6 | 25.8 | 602.7 | 1.0 | 98.2 | x |
| b-pinene | GC | 84.4 | 39.7 | 821.3 | 1.0 | 110.4 | x |
| toluene | GC | 48.8 | 27.4 | 820.0 | 2.3 | 76.5 | x |
| ethylbenzene | GC | 17.6 | 7.5 | 128.0 | 1.1 | 26.6 | x |

| | | | | | | | |
|---|---|---|---|---|---|---|---|
| m+p-xylene | GC | 50.9 | 11.8 | 471.0 | 2.0 | 92.9 | x |
| o-xylene | GC | 20.2 | 5.5 | 168.0 | 1.3 | 32.2 | x |
| $C_2HCl_3$ | GC | 0.7 | 0.5 | 3.0 | 0.1 | 0.6 | x |
| $C_2Cl_4$ | GC | 2.0 | 1.6 | 10.2 | 0.9 | 1.1 | x |
| $CH_3I$ | GC | 0.5 | 0.5 | 1.8 | 0.0 | 0.3 | |
| $CH_2Br_2$ | GC | 0.8 | 0.8 | 1.1 | 0.5 | 0.1 | |
| $CHBrCl_2$ | GC | 0.8 | 0.7 | 1.6 | 0.4 | 0.2 | |
| $CHBr_3$ | GC | 0.6 | 0.6 | 1.6 | 0.1 | 0.2 | |
| $MeONO_2$ | GC | 6.0 | 5.9 | 11.7 | 3.2 | 1.2 | x |
| $EtONO_2$ | GC | 2.7 | 2.4 | 11.8 | 1.1 | 1.4 | x |
| $i\text{-}PrONO_2$ | GC | 7.6 | 5.7 | 39.0 | 1.8 | 5.5 | x |
| $n\text{-}PrONO_2$ | GC | 1.8 | 1.3 | 10.1 | 0.4 | 1.3 | x |
| $2\text{-}BuONO_2$ | GC | 7.8 | 4.4 | 67.5 | 0.8 | 9.2 | x |
| $3\text{-}PenONO_2$ | GC | 1.9 | 0.9 | 19.9 | 0.1 | 2.6 | x |
| $2\text{-}PenONO_2$ | GC | 2.4 | 1.2 | 29.2 | 0.1 | 3.7 | x |
| methanol | PTRMS | 2780 | 2510 | 11400 | 660 | 1330 | |
| acetaldehyde | PTRMS | 380 | 320 | 2220 | 60 | 250 | |
| acetone | PTRMS | 1270 | 1100 | 3790 | 120 | 670 | x |
| acetic acid | PTRMS | 1060 | 1010 | 3800 | 50 | 540 | |
| MVK+MACR | PTRMS | 137 | 122 | 414 | 25 | 65 | |
| MEK | PTRMS | 130 | 100 | 670 | 20 | 91 | |
| monoterpenes | PTRMS | 160 | 130 | 680 | 50 | 92 | |
| benzene | PTRMS | 85 | 76 | 370 | 30 | 35 | x |
| toluene | PTRMS | 63 | 53 | 560 | 20 | 39 | |
| $C_8$ aromatics | PTRMS | 120 | 110 | 770 | 40 | 60 | |
| $C_9$ aromatics | PTRMS | 110 | 100 | 340 | 80 | 50 | |

**Table 2.** Summary of high O$_3$ events where at least 1 hour exceeded 70 ppb. Timing of the exceedance and the average peak concentration for both the Longs Peak and Trail Ridge monitoring sites are also included. Times shown are local.

| | ROMO-LP | | | ROMO-TR | |
|---|---|---|---|---|---|
| | hours exceeding 70 ppb | average for exceedance | i- to n-pentane | hours exceeding 70 ppb | average for exceedance |
| 7/22/2014 | 17:00-20:00 | 74 | 1.12 | 17:00-22:00 | 77 |
| 8/2/2014 | 17:00 | 71 | 1.05 | 20:00-21:00 | 71 |
| 8/8/2014 | 17:00 | 75 | 0.97 | – | – |
| 8/11/2014 | 15:00-16:00 | 73 | 1.44 | 17:00, 20:00, 21:00 | 70 |
| 8/12/2014 | 14:00-17:00 | 76 | 1.32 | 15:00-20:00 | 74 |
| 8/18/2014 | 17:00 | 74 | 0.97 | 20:00-21:00 | 72 |
| 8/23/2014 | 17:00 | 72 | 1.36 | – | – |
| 8/24/2014 | – | – | – | 11:00 | 70 |

**Table 3. Summary of the $O_3/RONO_2$ slopes (ppbv/ppbv), intercept values (ppbv), the range of $O_3$ attributable to each parent alkane, and the average $O_3$ attributable to each parent alkane based on the correlation in each of the $O_3$-alkyl nitrate scatter plots for the upslope events on August 8 and August 18. An orthogonal distance regression was used to determine the slope and intercept.**

| | $RONO_2$ | Slope ($O_3/RONO_2$) | Intercept (ppbv) | Attributed $O_3$ Range Min–Max (ppbv) | Attributed $O_3$ Range Ave (ppbv) |
|---|---|---|---|---|---|
| August 8 Event | EtONO2 | 2.9E+03 | 42 | 20 – 26 | 22 |
| 16:00-18:00 | i-PrONO2 | 7.9E+02 | 40 | 24 – 30 | 27 |
| (MST) | n-PrONO2 | 2.6E+03 | 44 | 17 – 24 | 20 |
| | 2-BuONO2 | 4.2E+02 | 43 | 22 – 28 | 24 |
| | 3-PenONO2 | 1.1E+03 | 46 | 16 – 21 | 18 |
| | 2-PenONO2 | 8.7E+02 | 45 | 18 – 23 | 20 |
| August 18 Event | EtONO2 | 2.7E+03 | 41 | 18 – 23 | 20 |
| 15:00-18:00 | i-PrONO2 | 6.3E+02 | 41 | 19 – 25 | 21 |
| (MST) | n-PrONO2 | 2.1E+03 | 44 | 13 – 21 | 16 |
| | 2-BuONO2 | 3.8E+02 | 42 | 18 – 26 | 21 |
| | 3-PenONO2 | 1.0E+03 | 45 | 14 – 20 | 17 |
| | 2-PenONO2 | 7.3E+02 | 45 | 16 – 21 | 18 |

**Table 4.** Percentage contributions of the light alkanes to the alkane OHR during the oil and gas influenced events at ROMO during FRAPPÉ (Total alkane OHR = 0.62 $s^{-1}$) and from measurements at the BAO and PAO ([1]Swarthout et al. (2013), [2]McDuffie et al. (2016), [3]Abeleria et al. (2017), [4]Halliday et al. (2016)).

| | ROMO Summer 2014 | BAO Winter 2011[1] | BAO Summer 2012[2] | BAO Spring 2015[3] | BAO Summer 2015[3] | PAO [5] Summer 2014[4] |
|---|---|---|---|---|---|---|
| ethane | 8 | 8 | 6 | 11 | 10 | 7 |
| propane | 26 | 27 | 22 | 30 | 19 | 22 |
| i-butane | 9 | 9 | 9 | 6 | 7 | 9 |
| n-butane | 24 | 25 | 24 | 15 | 19 | 23 |
| i-pentane | 13 | 11 | 14 | 15 | 19 | 13 |
| n-pentane | 13 | 12 | 16 | 15 | 19 | 15 |
| $C_6$-$C_9$ | 8 | 7 | 10 | 8 | 7 | 11 |
| $\sum OHR_{alkane}$ ($s^{-1}$) | 0.61 | 1.59 | 1.52 | 0.66 | 1.04 | 3.07 |