# Peer review of "Volatile Organic Compounds and Ozone in Rocky Mountain National Park during FRAPPÉ"

_Atmospheric Chemistry and Physics, 2018_

## Referee Comment (RC1) · Anonymous Referee #1 · 11 Jun 2018

**Overall:** This paper reports very useful observations and contains a lot of useful analysis. However, certain portions of the discussion (principally in Section 4, but also 1 paragraph in Section 3) are poorly supported at best. The conclusions drawn by those sections may well be correct, but the manuscript (MS) does not demonstrate them.

My hope is that the authors can rewrite the portions of the MS that are not adequately supported and address some other issues I catalog below so that the MS can be published in ACP.

Furthermore, the paper reports that oil and gas (OG) VOC contributes to about 20 ppbv of O3 in Rocky Mountain (ROMO). This is very hard to reconcile with the modeling

[Figure]

results of McDuffie (2016), which are reporting that at BAO, peak ozone concentrations would only be 3 ppbv lower if OG VOC were not present. This paper is cited in the MS, but this critical difference is not mentioned. It does not seem likely that both of these results can be correct: OG VOC only adds about 3 ppbv at BAO but 20 ppbv at ROMO. If there is an explanation for this, or if there is some other explanation for this discrepancy, it should be discussed. If the authors are not aware of an explanation for the discrepancy, that should be acknowledged.

**Section 4.2** First, the PAN & PPN discussion in the 3rd paragraph is difficult to follow, seems tendentious, and it's not clear to me that it adds to the analysis. PPN was not measured at ROMO, but the arguments here seem to rely analysis in a paper by Lindaas (submitted) that says that high PPN/PAN ratios indicate substantial contribution to PAN and PPN from OG alkanes. If these arguments are important to the overall analysis, they should be clarified and simplified. It appears to me that the key result is simply that previous work studying the front range has demonstrated that OG alkanes are a key precursor to PAN and PPN in this area; the measurements of PAN and ROMO don't seem to add much to this.

**Page 12-13** – analysis based on alkyl nitrates. These two paragraphs, which lead to a key finding of the MS ( 20 ppbv O3 from OG VOC) are at best inadequately supported. The logic of many of the steps in this analysis is not transparent: simply put, I can't see why its true. These paragraphs are also very, very sparse on citations, so more support is certainly needed.

P12 lines 11-14. I'm afraid I am not familiar with using alkyl nitrate formation as a proxy for O3 production in this way. Please provide citations to papers that explain the logic of this approach, or do so here (ie with Chemical Equations, etc).

Lines 14-16 "For the August 8 and 18 events, the overall air mass composition (Figure 11) and photochemical age (Figure 12) indicate that the source region emissions and processing times in the Front Range were comparable and these assumptions are

valid." The assumptions are the three stated on lines 11-14. First, Figs 11 and 12 don't tell us anything about co-location of NOx and VOC. Second, Fig 11 does not allow us to compare anything about Aug 8 and 18, because the data for those two days is averaged on Fig 11. Fig 12 does show that the photochemical ages appear to be quite similar for the high ozone hours (Fig 12 is hourly data?) on 8/8 and 8/18, and on the order of 15 hours. But that is not direct support for any of the three assumptions.

In short, this paragraph does not help me understand why this approach makes sense if the 3 assumptions are valid, nor why the data shown in Figs 11 and 12 supports the validity of those 3 assumption.

Page 13 lines 3 – 8. Again, this simple correlation of alkyl nitrates to O3 as a way of attributing O3 to alkenes doesn't seem obvious to me, and the MS just does not explain why this is logical.

The agreement in results when this is done for various alkyl nitrates is interesting, but not convincing, since this analysis does not tell us anything about the contribution from other VOC (non-alkanes) that are simultaneously contributing to O3.

**In short, this analysis needs to be justified.**

Page 13 lines 8-9 – consistency with Cheadle 2017. **Be careful here.** First, Cheadle is claiming that on the day when Cheadle suggests the highest contribution of OG emissions to measured O3 (8/13), NOx was also attributable to OG. Therefore, Cheadle is attributing up to 30 ppb O3 to all OG emissions, not just OG VOC. Second, the conditions on the high O3 days were light winds. In these conditions, emissions can accumulate to very high levels in air over Weld Cty (where Cheadle measured). For example on 8/3 Platteville flasks measured 50 – 150 ppb ethane – far higher than ethane at ROMO on 8/8 or 8/18. Therefore, I don't agree that Cheadle's results are really consistent with what the MS is claiming about O3 at ROMO attributable to OG VOC.

Page 13 line 16 – again, Fig 11 simply does not show how similar airmasses were on

[Figure]

Aug 8 and 18 because the data for those two days is avgd. in the figure.

There is no question, based on the results shown in the MS, that OG contributes to O3 at ROMO. However, the analysis used in the MS to estimate that 20 ppbv of the peak O3 on these two days is attributable to OG VOC seems very questionable.

**Section 3** Page 8-9, paragraph beginning on p 8 line 19, and Figure 6 This paragraph makes a number of unsupported statements – which may very well be true – and is poorly organized. Figure 6 does not well support the statements and has issues itself.

As it stands now, the statements that NMHC levels at ROMO "can be comparable . . . to other [sic] urban/industrial areas" and "Observed mixing ratios during afternoons with upslope flow can be similar to those observed simultaneously in the Front Range" are not supported.

These statements appear to be the main point of the paragraph, but they are not supported by Fig 6, since Fig 6 clearly shows significantly lower MRs for ROMO than everywhere else. I suspect that this has a lot to do with the averaging, so I don't doubt the statements above, but again, the statements are not supported.

- First and foremost with the figure, what data are being compared. I presume that the transect whole air samples are all collected during the day (I don't think this is stated either way), while the ROMO data are averages / medians for 24-hour data? Since Fig 3 appears to show that at ROMO the spikes in NMHC are all daytime only when upslope is happening, comparing the 24 ROMO average to daytime-only (?) transect data arguably artificially decreases the ROMO data in this sense.

- Then, the figure compares these values to PAO data and BAO data. But, those are probably 24hour averages (???) and may well include high values measured at night with low boundary layers (?). I'm speculating, but there is no discussion of any of this. How comparable are these values?

The simple way to handle this might be to compare values at ROMO during upslope

events to values at PAO, or during afternoons at ROMO. But right now, the statements are not supported.

There are other issues with the paragraph and figure that need to be addressed.

- BAO is first introduced here, but the acronym is not defined, and no context about the site (ie, "a site in the NFRMA somewhat more removed from OG production than PAO") is provided.

- The fact that ethyne and benzene are similar at ROMO and during the transects, and they are far more similar to PAO and BAO levels than the alkanes, needs some discussion. It's just mentioned in passing. ???

- For the figure:

o Define the limits of the boxes and whiskers for the data from this study

o Define the error bars for the Baker 2008 data

o Ideally, draw vertical lines or something like that between each set of symbols (that is, a line between the ethane and propane symbols, a line btwn the propane and i-C4 symbols, a line btwn i- and n-C4 symbols, etc.) As it is now it's really hard to read the graph.

**Minor Issues**

- It's really helpful for future readers if you put the dates of the observations (months and the year) in the abstract

- P4 line 6 – what time of day were the canisters collected?

- P4, lines 14-16 – this is not very clear or helpful. First, I think it means that the analytical system was calibrated for multiple species with a whole air standard which was run every 10 samples. Second, the precisions listed are very good (1%!) . . . but at what mole fraction? Comparable to the mole fractions measured at ROMO? A specific

reference here is needed, even it it is one of the 3 papers listed in the pvs sentence.

- P6, lines 16-19. This passage about the contents of propane LPG cylinders is poorly drafted and mildly inaccurate (it appears to carry over an error from Wikipedia about the specifics of the hd-5 standard). The "mild error:" HD-5 must be >90% propane, <5% propene, and <2.5% C4+, according to astm 1835 (those are liquid vol %s). There is no limit of ethane except the 10% upper limit (since the gas must be 90% propane). This passage needs proper references.

- Page 7 – top – headings. Having two headings here is superfluous... I suggest you replace the two headings with "3. Temporal and Spatial Trends in Volatile Organic Compounds"

- Page 7, line 17-18 – significant MeOH from oil and gas has been observed in Utah by NOAA. Is this not the case in the DJ in the summertime? I can believe it since I think it's main use for OG is inhibition of hydrate formation in cold weather. But I think that would be valuable to note; it should not be assumed that OG is not part of the source mix for MeOH at ROMO.

- Page 7 lines 19-20. It won't be obvious to some readers what "hourly 2-day ensemble back trajectories" are. (I don't know in what sense they are "ensembles"). How about "two-day back trajectories were calculated for each hour of the study"

- Figure 4. The units are not clear. Is this the % of trajectories that include (go through) each pixel, or that end in each pixel? Clarify. The term ORT seems to imply the former to this reader, but the units at the top of the color bar says "% ends." Also the units are cut off on the figure. %ends / g???

---

## Referee Comment (RC2) · Anonymous Referee #2 · 12 Jun 2018

General Comments:

This manuscript contributes a valuable documentation to ozone and VOC data for the Rocky Mountain National Park (ROMO) during the FRAPPE campaign time period of summer 2014. I recommend that it be published in ACP.

-I agree with some of the concerns of the first reviewer - plus the following.

For future search/reference to this data, please add a date reference such as "summer 2014" to the abstract.

I recommend that the paper recognizes the main FRAPPE "infrastructure" funding

[Figure]

sources ie NCAR and the Colorado Department of Public Health and Environment (CDPHE), as well as NASA - if only for archive maintainance.

Figure 1 Maybe there could be a more contextual discussion of ozone for the FRAPPE campaign time period compared to the full time series shown in this figure?

Figure 6 Maybe add PAN?

[Figure]

---

## Author Comment (AC1) · 19 Sep 2018

We appreciate the constructive comments on our manuscript provided by the referees. We have taken these into consideration and modified the paper in response. Overall, the suggestions have made this a stronger paper. The responses are explained in our response to reviewer comments below with all of our responses are in red type. Changes or additions to the manuscript are *red and italic*. At the end of the responses (pg 9) the complete manuscript is provided with all changes marked since the initial submission.

**Overall:** This paper reports very useful observations and contains a lot of useful analysis. However, certain portions of the discussion (principally in Section 4, but also 1 paragraph in Section 3) are poorly supported at best. The conclusions drawn by those sections may well be correct, but the manuscript (MS) does not demonstrate them. My hope is that the authors can rewrite the portions of the MS that are not adequately supported and address some other issues I catalog below so that the MS can be published in ACP.

Furthermore, the paper reports that oil and gas (OG) VOC contributes to about 20 ppbv of O3 in Rocky Mountain (ROMO). This is very hard to reconcile with the modeling results of McDuffie (2016), which are reporting that at BAO, peak ozone concentrations would only be 3 ppbv lower if OG VOC were not present. This paper is cited in the MS, but this critical difference is not mentioned. It does not seem likely that both of these results can be correct: OG VOC only adds about 3 ppbv at BAO but 20 ppbv at ROMO. If there is an explanation for this, or if there is some other explanation for this discrepancy, it should be discussed. If the authors are not aware of an explanation for the discrepancy, that should be acknowledged.

It is stated in the paper that we are investigating the influence of O&G emissions for specific, well characterized upslope events. Our approach and focus differs substantially from the McDuffie et al. work. Their study reports average O&G contributions, and they use VOC abundances from 2012 with $NO_x$ and $O_3$ measurements from 2014. This paper is highly reliant on a set of model experiments. Co-authors on this paper, are also co-authors on the McDuffie paper. We are not criticizing the approach in McDuffie et al, but rather acknowledging the limitation of the approach and the strength in approaching this problem using multiple approaches. Since the McDuffie et al., work, there have been many other ozone-focused papers. Most relevant, Cheadle et al. [2017] estimate that on individual days, oil and gas O3 precursors can contribute up to 30 ppbv of ozone. The reviewer mentions this paper later and points out that this 30 ppbv is associated with both VOC and NOx precursors. Also relevant is Zaragoza et al. [2017]. This paper shows that on days with large ozone production, anthropogenic VOCs dominate the photochemistry. The PAN ratios on these days indicate that alkane chemistry is important.

A close examination of Figures 3, 12, 1S and 2S shows that impacts during upslope events can be quite pronounced, while O&G impacts averaged over the whole summer season are significantly lower. Therefore, we use an *event based approach* focused on upslope events from the NFRMA to show that Front Range emissions, in particular O&G coupled with urban emissions, are driving the ozone exceedances in ROMO. We have added text to the manuscript to clarify this difference. In addition, we have also de-emphasized the 20 ppb increase in $O_3$ at ROMO is just from oil and gas due to the uncertainty in the calculations associated with the method we have used. In particular we have added that other species like OVOCs may be contributing to this increase in $O_3$. However, this is conservative because many of the OVOCs (MEK, acetone and acetaldehyde) can be produced from alkane oxidation.

Cheadle LC, Oltmans SJ, Petron G, Schnell RC, Mattson EJ, Herndon SC, et al.. **Surface ozone in the Colorado northern Front Range and the influence of oil and gas development during FRAPPE/DISCOVER-AQ in summer 2014**. Elem Sci Anth. 2017;5:61. DOI: http://doi.org/10.1525/elementa.254

Zaragoza, J., Callahan, S., McDuffie, E. E., Kirkland, J., Brophy, P., Durrett, L., et al. (2017). **Observations of acyl peroxy nitrates during the Front Range Air Pollution and Photochemistry Éxperiment (FRAPPÉ).** Journal of Geophysical Research: Atmospheres, 122. https://doi.org/10.1002/2017JD027337

**Section 4.2** First, the PAN & PPN discussion in the 3rd paragraph is difficult to follow, seems tendentious, and it's not clear to me that it adds to the analysis. PPN was not measured at ROMO, but the arguments here seem to rely analysis in a paper by Lindaas (submitted) that says that high PPN/PAN ratios indicate substantial contribution to PAN and PPN from OG alkanes. If these arguments are important to the overall analysis, they should be clarified and simplified. It appears to me that the key result is simply that previous work studying the front range has demonstrated that OG alkanes are a key precursor to PAN and PPN in this area; the measurements of PAN and ROMO don't seem to add much to this.

We have clarified and simplified this discussion to focus on what the presence of PAN indicates at ROMO (PAN likely formed from OG alkanes).

**Page 12-13** – analysis based on alkyl nitrates. These two paragraphs, which lead to a key finding of the MS ( 20 ppbv O3 from OG VOC) are at best inadequately supported. The logic of many of the steps in this analysis is not transparent: simply put, I can't see why its true. These paragraphs are also very, very sparse on citations, so more support is certainly needed.
The following has been added to the manuscript, which highlights the point that a limited number of similar types of analysis are reported in the literature.

*As noted in Rosen et al. (2004), only a limited number of studies report the correlation of $O_3$ or $O_x$ with alkyl nitrates, with half of the studies using the sum of alkyl nitrates ($\sum$ANs) because a non-selective technique was used that did not allow for individual speciation of the alkyl nitrates (Flocke et al., 1991; Neuman et al., 2012; O'Brien et al., 1995; Perring et al., 2013; Roberts et al., 1996; Rosen et al., 2004). Flocke et al. (1991) reported the correlation of $\Box C_2$–$C_5$ straight chain alkane-derived nitrates plus a single $C_4$ branched chain alkane-derived nitrate with $O_x$ measured at Shauinsland, Germany.*

P12 lines 11-14. I'm afraid I am not familiar with using alkyl nitrate formation as a proxy for O3 production in this way. Please provide citations to papers that explain the logic of this approach, or do so here (ie with Chemical Equations, etc).
The following text has been added to better illustrate the utility of using alkyl nitrates as a proxy for ozone production:

*Because the production of $O_3$ and alkyl nitrates is tied to their common precursor, the alkyl peroxy radical ($RO_2$), correlation between $O_3$ and the $C_2$-$C_5$ alkyl nitrates is expected. As a first step, the y-intercept is used to estimate the background $O_3$ value for the event period prior to the start of the upslope event. As shown in Table 3, each of the individual alkyl nitrates gives a slightly different background $O_3$ value (y-intercept); however, they are all within a few ppbv of each other, providing a reasonable estimate of the background $O_3$ during that time period. The values reported for this work are also in the range of values reported by McDuffie et al. (2016) for BAO during summer 2014. The main source of the $C_2$-$C_5$ alkyl nitrates is the photochemical production from their parent alkane precursors, as outlined in the following simplified reaction scheme:*

$$RH + OH \rightarrow R + H_2O \qquad (R1)$$
$$R + O_2 \rightarrow RO_2 \qquad (R2)$$
$$RO_2 + NO \rightarrow RO + NO_2 \qquad (R3a)$$
$$RO_2 + NO + M \rightarrow RONO_2 + M \qquad (R3b)$$
$$NO_2 + h\upsilon \rightarrow NO + O \qquad (R4)$$
$$O + O_2 + M \rightarrow O_3 + M \qquad (R5)$$

*As shown by Reactions 1-5, alkyl nitrates share a common photochemical production mechanism with $O_3$. However, $O_3$ formation results from the photolysis of $NO_2$ (R4) whereas the formation of alkyl nitrates serve as a sink for $NO_x$, RO and $RO_2$, which affects $O_3$ production efficiency (Atkinson et al., 1982; Ranschaert et al., 2000; Russo et al., 2010b). Nonetheless, based on these reactions, it is expected that under a wide range of conditions $O_3$ and alkyl nitrates should be correlated as they are*

*produced concurrently in the atmosphere (e.g. Abeleira et al., 2018; Day et al., 2003; Flocke et al., 1991; Perring et al., 2013; Rosen et al., 2004; Russo et al., 2010b).*

Lines 14-16 "For the August 8 and 18 events, the overall air mass composition (Figure 11) and photochemical age (Figure 12)
5 indicate that the source region emissions and processing times in the Front Range were comparable and these assumptions are valid." The assumptions are the three stated on lines 11-14. First, Figs 11 and 12 don't tell us anything about co-location of NOx and VOC. Second, Fig 11 does not allow us to compare anything about Aug 8 and 18, because the data for those two days is averaged on Fig 11. Fig 12 does show that the photochemical ages appear to be quite similar for the high ozone hours (Fig 12 is hourly data?) on 8/8 and 8/18, and on the order of 15 hours. But that is not direct support for any of the three
10 assumptions. In short, this paragraph does not help me understand why this approach makes sense if the 3 assumptions are valid, nor why the data shown in Figs 11 and 12 supports the validity of those 3 assumption.

The wrong figure was cited – it should have been Figure 14 (now Figure 12). This has been corrected. Additionally, the following text has been added to detail that these assumptions were met:

15 *Analogous to PAN, alkyl nitrates are photochemically produced simultaneously with ozone in the atmosphere. Here, we use coincident observations of alkyl nitrates during these elevated $O_3$ periods to estimate the contribution of the light alkanes from oil and gas emissions to $O_3$ production. Alkyl nitrate formation can be used as a proxy for $O_3$ production if we make the following assumptions. First, their photochemical production is rapid compared to removal processes and mixing across gradients. The rate of alkyl nitrate formation is tied to*
20 *its parent alkane's concentration and reaction rate with the hydroxyl radical (OH), which is the hydroxyl radical reactivity (R1), as this is the rate limiting step in alkyl peroxy radical ($RO_2$) formation. In NFRMA areas that are influenced by oil and gas emissions, the hydroxyl radical reactivity values for the alkyl nitrate parent alkanes are high (Figure 11), resulting in rapid production compared to their removal rates. Because of their relatively long lifetimes ($\sim \leq 10$ days for summer), the $C_2$-$C_5$ alkyl nitrates can be transported long distances and serve as a*
25 *temporary reservoir for $NO_x$, ultimately leading to $O_3$ production in downwind or remote regions (e.g., Clemitshaw et al., 1997; Flocke et al., 1998; Roberts et al., 1998). For the upslope events on August 8 and 18, the assumption that photochemical production of the alkyl nitrates is rapid compared to removal processes is supported based on the rates of formation (hours) versus their removal (days). Second, the precursor compounds (VOCs and $NO_x$) both have sources in Weld County, where significant oil and gas development exists $NO_x$ was elevated during both*
30 *events (not shown) and the 2014 National Emissions Inventory (US EPA, 2017) suggests Weld County has the largest emissions of $NO_x$ and VOCs in the state, and so it is reasonable to assume that emissions are collocated. The collocation of sources indicates these air masses likely contained sufficient $NO_x$ for alkyl nitrate and $O_3$ formation and the excess $O_3$ is clearly related to VOC and $NO_x$ emissions from the oil and gas region. There could be $NO_x$ additions to the air mass as it moves west, but this does not violate the assumption because of the timescale*
35 *for the transport, which is relatively fast. Third, transport time from the source region (NFRMA) to the park is rapid (on the order of hours). This assumption is validated by the time series plots shown in Figure 13, where during the upslope events, rapid increases in mixing ratios are observed within an hour of when the wind shifts to an upslope direction.*

*In general, this approach is similar to estimating $O_3$ production efficiency and background $O_3$ using $O_x/NO_z$*
40 *but instead we use the individual alkyl nitrates in $NO_z$ because of the abundance of alkanes emitted from oil and gas operations. Moreover, we can better apportion the photochemical processing from the source region emissions because we are using an ensemble of individual compounds that have known rate constants, branching ratios and processing times, as opposed to using a bulk parameter such as a $NO_z$, where the composition is not accurately known. For the oil and gas influenced upslope events at ROMO, PAN, $O_3$, $NO_x$, and alkyl nitrates all show*
45 *coincident increases (Figure 3; Benedict et al., 2018). For the August 8 and 18 events, the overall air mass composition (Figures 3, 12, 1S and 2S) and photochemical age (Figure 13) indicate that the source region emissions and processing times in the Front Range were comparable and these assumptions are valid.*

Page 13 lines 3 – 8. Again, this simple correlation of alkyl nitrates to O3 as a way of attributing O3 to alkenes doesn't seem obvious to me, and the MS just does not explain why this is logical. The agreement in results when this is done for various alkyl nitrates is interesting, but not convincing, since this analysis does not tell us anything about the contribution from other VOC (non-alkanes) that are simultaneously contributing to O3.

**In short, this analysis needs to be justified.**

We have addressed these concerns in the previous responses by including more detail describing how the excess ozone was calculated from the $O_3/RONO_2$ plots.

Page 13 lines 8-9 – consistency with Cheadle 2017. **Be careful here.** First, Cheadle is claiming that on the day when Cheadle suggests the highest contribution of OG emissions to measured O3 (8/13), NOx was also attributable to OG. Therefore, Cheadle is attributing up to 30 ppb O3 to all OG emissions, not just OG VOC. Second, the conditions on the high O3 days were light winds. In these conditions, emissions can
accumulate to very high levels in air over Weld Cty (where Cheadle measured). For example
on 8/3 Platteville flasks measured 50 – 150 ppb ethane – far higher than ethane at ROMO on 8/8 or 8/18. Therefore, I don't agree that Cheadle's results are really consistent with what the MS is claiming about O3 at ROMO attributable to OG VOC.

We agree that different weather conditions existed for the determination Cheadle et al. made compared with those we are making. The stagnation conditions experienced on the day Cheadle et al. used to make their estimate of OG emissions to $O_3$ are often the conditions that lead to elevated $O_3$ in ROMO if the elevated concentrations of VOCs, $NO_x$, and $O_3$ are transported to ROMO from the Front Range. The air masses that we measured during upslope events originated on the Front Range, and the mix of VOCs suggest air masses highly influenced by O&G with elevated $O_3$. It is also worth noting that the methods for determining the increase in $O_3$ from O&G are quite different. We have added the following clarification to the text that regarding NOx.

*Second, the precursor compounds (VOCs and $NO_x$) both have sources in Weld County, where significant oil and gas development exists $NO_x$ was elevated during both events (not shown) and the 2014 National Emissions Inventory (US EPA, 2017) suggests Weld County has the largest emissions of $NO_x$ and VOCs in the state, and so it is reasonable to assume that emissions are collocated. The collocation of sources indicates these air masses likely contained sufficient $NO_x$ for alkyl nitrate and $O_3$ formation and the excess $O_3$ is clearly related to VOC and $NO_x$ emissions from the oil and gas region.*

Page 13 line 16 – again, Fig 11 simply does not show how similar airmasses were on Aug 8 and 18 because the data for those two days is avgd. in the figure. There is no question, based on the results shown in the MS, that OG contributes to O3 at ROMO. However, the analysis used in the MS to estimate that 20 ppbv of the peak O3 on these two days is attributable to OG VOC seems very questionable.

Figure 14 (now Figure 12) should be referenced here instead of Figure 11. This was overlooked when the order of figures was rearranged before submission. We have also added a reference to Figure 3 which shows that the mix of species is similar on both days. Regarding the estimate of 20 ppb O3, we have expanded our attribution to include other VOCs (i.e. OVOCs) and noted the elevated NOx present. We have also added that this is an upper limit of the oil and gas contribution to elevated O3 on these days.

*These values represent an upper limit on the excess $O_3$ produced from NFRMA air masses transported to ROMO that were dominated by oil and gas emissions. Other compounds produced photochemically in situ (e.g., PAN and acetone) provide similar values to the alkyl nitrates for the attributed $O_3$ (not shown). The alkyl nitrate correlation method does not allow us to*

*explicitly separate the impact of NO$_x$, additional OVOCs or other species such as alkenes (Figure 11), which all likely contribute to ozone production.*

**Section 3** Page 8-9, paragraph beginning on p 8 line 19, and Figure 6 This paragraph makes a number of unsupported statements – which may very well be true – and is poorly organized. Figure 6 does not well support the statements and has issues itself. As it stands now, the statements that NMHC levels at ROMO "can be comparable . . . to other [sic] urban/industrial areas" and "Observed mixing ratios during afternoons with
upslope flow can be similar to those observed simultaneously in the Front Range" are
not supported. These statements appear to be the main point of the paragraph, but they are not supported
by Fig 6, since Fig 6 clearly shows significantly lower MRs for ROMO than everywhere else. I suspect that this has a lot to do with the averaging, so I don't doubt the statements above, but again, the statements are not supported.
We have reorganized this paragraph to better convey the main point – that peak values observed at ROMO are at the levels observed in other areas. We readily acknowledge that the majority of observed values at ROMO are lower. We have added citations supporting lower mixing ratios at remote sites (Jobson et al., 1994; Rindsland et al., 2002; Rudolph, 1995; Simpson et al., 2012). We have clarified this in the text with the following:

*At their peak values, NMHC mixing ratios observed at ROMO can be of comparable magnitude to urban/industrial regions, further suggesting a significant impact of the polluted NFRMA on ROMO during specific events. The range of mixing ratios observed at ROMO overlaps the range of observations compiled by Baker et al. (2008) from 28 cities across the United States.*

- First and foremost with the figure, what data are being compared. I presume that the transect whole air samples are all collected during the day (I don't think this is stated either way), while the ROMO data are averages / medians for 24-hour data? Since Fig 3 appears to show that at ROMO the spikes in NMHC are all daytime only when upslope is happening, comparing the 24 ROMO average to daytime-only (?) transect
data arguably artificially decreases the ROMO data in this sense.
We have added the following sentence to explain the high transect mixing ratios in the transect samples compared to ROMO:

*The transect samples were collected next to the road and only during daytime hours, when upslope flow from the east is prevalent.*

- Then, the figure compares these values to PAO data and BAO data. But, those are probably 24hour averages (???) and may well include high values measured at night with low boundary layers (?). I'm speculating, but there is no discussion of any of this. How comparable are these values? The simple way to handle this might be to compare values at ROMO during upslope events to values at PAO, or during afternoons at ROMO. But right now, the statements are not supported.
Most of the data presented are from datasets that cover 24 hours of measurements. The exceptions to this are the transect samples and the Baker et al. data which represent a range of observations from 28 different cities. We have added the time of data collection for each study to the caption in Figure 6 to clarify the comparison across sites.

There are other issues with the paragraph and figure that need to be addressed.
- BAO is first introduced here, but the acronym is not defined, and no context about the site (ie, "a site in the NFRMA somewhat more removed from OG production than PAO") is provided.
We have added a description of BAO.

- The fact that ethyne and benzene are similar at ROMO and during the transects, and they are far more similar to PAO and BAO levels than the alkanes, needs some discussion. It's just mentioned in passing. ???

The mixing ratios of benzene and ethyne likely reflect background values. These species have longer atmospheric lifetimes and are relatively well mixed unless measurements are made in a plume. We have added some discussion of these values likely representing background to the manuscript.

*The ROMO and transect ethyne mixing ratios presented in this figure likely represent background as there is significant overlap for all sites with the range of means from 28 U.S. cities. Benzene mixing ratios also suggest background values for the ROMO and transect samples while slightly elevated mixing ratios at BAO and PAO indicate additional sources, likely from oil and gas (Halliday et al., 2016).*

- For the figure:
o Define the limits of the boxes and whiskers for the data from this study
o Define the error bars for the Baker 2008 data
o Ideally, draw vertical lines or something like that between each set of symbols (that is, a line between the ethane and propane symbols, a line btwn the propane and i-C4 symbols, a line btwn i- and n-C4 symbols, etc.) As it is now it's really hard to read the graph.

We have made the suggested changes.

**Minor Issues**
- It's really helpful for future readers if you put the dates of the observations (months and the year) in the abstract

We have added the months and year of measurement to the abstract.

- P4 line 6 – what time of day were the canisters collected?

We have added that the canister samples were generally collected between 13:00-18:00.

- P4, lines 14-16 – this is not very clear or helpful. First, I think it means that the analytical system was calibrated for multiple species with a whole air standard which was run every 10 samples. Second, the precisions listed are very good (1%!) . . . but at what mole fraction? Comparable to the mole fractions measured at ROMO? A specific reference here is needed, even it it is one of the 3 papers listed in the pvs sentence.

We have added the following information to the manuscript to clarify the range of standards analyzed:

*The whole air working standards employed for this work have mixing ratios representative of clean free tropospheric air (NMHCs mixing ratios on the order of ~10-100 pptv) and suburban air (NMHC mixing ratios on the order of ~1-2 ppbv), thus bracketing the low and high ranges for the measurements at ROMO (e.g. Russo et al., 2010a).*

- P6, lines 16-19. This passage about the contents of propane LPG cylinders is poorly drafted and mildly inaccurate (it appears to carry over an error from Wikipedia about the specifics of the hd-5 standard). The "mild error:" HD-5 must be >90% propane, <5% propene, and <2.5% C4+, according to astm 1835 (those are liquid vol %s). There is no limit of ethane except the 10% upper limit (since the gas must be 90% propane).
This passage needs proper references.

We have updated this section with the correct information and have added the following reference: ASTM Standard D1835-16, 2016, "Standard Specification for Liquefied Petroleum (LP) Gases," ASTM International, West Conshohocken, PA, 2016, DOI: 10.1520/ D1835-16, www.astm.org.

- Page 7 – top – headings. Having two headings here is superfluous... I suggest you replace the two headings with "3. Temporal and Spatial Trends in Volatile Organic Compounds"

We agree and we have made the change as suggested.

- Page 7, line 17-18 – significant MeOH from oil and gas has been observed in Utah by NOAA. Is this not the case in the DJ in the summertime? I can believe it since I think it's main use for OG is inhibition of hydrate formation in cold weather. But I think that would be valuable to note; it should not be assumed that OG is not part of the source mix for MeOH at ROMO.

We have added this possibility to the text.  The text now reads:

*Methanol has vegetative sources and potential oil and gas sources in some basins where it is used during cold weather to inhibit equipment freezing (Warneke et al., 2014), but, more importantly for the Front Range, in summer this trace gas also comes from animal agriculture and confined animal feeding operations (CAFOs)(Sun et al., 2008).  Given the co-location of agricultural and oil and gas related sources in NE Colorado, it is difficult to ascertain the magnitude of each source, but it is likely that agriculture and CAFOS would be the larger source in the NFRMA and surrounding areas during the summer.*

- Page 7 lines 19-20. It won't be obvious to some readers what "hourly 2-day ensemble back trajectories" are. (I don't know in what sense they are "ensembles"). How about "two-day back trajectories were calculated for each hour of the study"

We have clarified this. It now reads: *To better understand VOC influence from these sources, air mass transport patterns were examined with back trajectories started hourly at 10 m above ground with previous air mass positions traced backward in time for two days using the Hybrid Single-Particle Lagrangian Integrated Trajectory (HYSPLIT) model ver 4.9.*

- Figure 4. The units are not clear. Is this the % of trajectories that include (go through) each pixel, or that end in each pixel? Clarify. The term ORT seems to imply the former to this reader, but the units at the top of the color bar says "% ends." Also the units are cut off on the figure. %ends / g???

We have added the following information to the caption to clarify the units.  *"Units are the percent of total endpoints in each grid cell.  Endpoints are upwind air mass locations calculated every hour for two days back in time for every trajectory that arrived at the receptor during the duration of the study period."*

Reviewer 2:

General Comments:

This manuscript contributes a valuable documentation to ozone and VOC data for the Rocky Mountain National Park (ROMO) during the FRAPPE campaign time period of summer 2014. I recommend that it be published in ACP.

-I agree with some of the concerns of the first reviewer - plus the following.

For future search/reference to this data, please add a date reference such as "summer 2014" to the abstract.

We have added the months and year of measurement to the abstract.

I recommend that the paper recognizes the main FRAPPE "infrastructure" funding sources ie NCAR and the Colorado Department of Public Health and Environment (CDPHE), as well as NASA - if only for archive maintainance.

We have added recognition of the FRAPPE team and funding sources to the acknowledgements section.

Figure 1 Maybe there could be a more contextual discussion of ozone for the FRAPPE campaign time period compared to the full time series shown in this figure?

We have added several sentences to address the levels of $O_3$ during the study compared to the record.  To the introduction we have added:

*In general, $O_3$ mixing ratios were lower during the summer of 2014 than in the preceding four years; specifically, the peak $O_3$ mixing ratios were lower and there were fewer hours when mixing ratios exceeded 70 ppb. Weather conditions, including greater rainfall and cloud cover than is typical during the summer months, contributed to lower $O_3$ levels in ROMO and on the Front Range (Cheadle et al., 2017; McDuffie et al., 2016).*

Figure 6 Maybe add PAN?

Because PAN data are not available for most of the other sites and studies included in the figure, we did not add PAN to Figure 6.

[revised manuscript text omitted]

**Oil & Gas**

Aug 8 & Aug 18

i/n-pentane <1

Total -OHR=1.26 s$^{-1}$

Alkanes 40%

Alkenes 12%

Ethyne 0.4%

Isoprene 7%

Aromatics 1.6%

OVOCs 39%

**Urban + Oil & Gas**

July 22, Aug 11, & Aug 12

i/n-pentane >1

Total VOC-OHR=0.825 s$^{-1}$

Alkanes 20%

OVOCs 40%

Alkenes 17%

Ethyne 0.7%

Isoprene 21%

Aromatics 1.1%

**Background**

July 21 & Aug 16

Total VOC-OHR=0.99 s$^{-1}$

Ethyne 0.1%

Alkenes 9.1%

Alkanes 1.8%

Isoprene 74%

OVOCs 14%

Aromatics 0.2%

**Stratosphere Influenced**

Aug 23

Total VOC-OHR=0.16 s$^{-1}$

Alkanes 14%

Alkenes 22%

Ethyne 1.9%

Aromatics 1.1%

Isoprene 2.3%

OVOCs 59%

[revised manuscript text omitted]

---

## Referee Report (RR1)

First, I apologize for completing this review late.

The manuscript (MS) has been significantly improved compared to the previous version.  In several places the authors have greatly clarified arguments which I previously had trouble following (or couldn't follow at all).   As stated before, the measurements and analysis described in the MS are very interesting and will contribute to our understanding of pollution events in this region, so this MS should be published.

There are, however, several remaining issues.  I first list two issues that need to be addressed, and then one suggestion.  I don't think it's necessary that a revised paper be reviewed again, but perhaps these issues are complicated enough that the Editor will prefer to have the reviewers take another look.  (In that case I will be happy to help out and I promise to be much faster!)

1. In the abstract of the previous version, the MS attributed ~20 ppb O3 during a couple high ozone events to oil and gas VOC.  The abstract of the current version moves away from these statements, attributing the same ~20 ppb O3 to "Front Range sources."  The new formulation in the abstract is a much better description of what the data analysis demonstrates.

Unfortunately, the text of the MS describing the analysis that derives the ~20 ppb O3 attribution still states, in several places, that the analysis can attribute ozone to more specific source categories such as "light alkanes from oil and gas."  I don't agree with that, and the MS is not internally consistent in this way.  The whole MS needs to be made consistent with the statement in the abstract.  The over-specific attribution statements need to be changed or removed.  Furthermore, the four page description of the analysis behind the source attribution - which is reorganized, with a lot of new text - needs to be shortened and substantially edited.  More description of the editorial problems with this section is below.

2.  In some places, the MS may still oversimplify the source attribution for the upslope events largely ascribed to OG emissions.  As I argue below, some of the evidence suggests that these O3 spikes can be described as mixed source events with a large OG signal, as opposed to "largely associated with VOC from OG," as described in the abstract.  Simply put, some of the data presented in the paper suggests to this reader that there could be a significant urban contribution to this O3. Other evidence / lines of analysis may complicate the story for those spikes.  But on balance, the story seems more complicated than simply saying that the OG signature is "dominant" for those two events.

I suggest that the authors handle this by either a) discussing why the indications of mixed sources (urban, etc.) should be discounted, or b) acknowledging that the evidence is somewhat mixed, and softening some of the statements that oil and gas emissions "dominated" these events.

Below are details on these two issues, followed by the Suggestion I mentioned above.
* * *
ISSUE 1.  **Nitrate analysis.**  The analysis described on pp 22-26 has substantially more explanation, but the discussion is difficult to follow in various places different conclusions are

attributed to this analysis.

In the abstract, the following statement is made:

"For the high ozone events most associated with emissions from oil and gas activities, we estimate that VOCs and NOx from sources along the Front Range contributed ~20 ppbv of additional ozone."

This is quite different than the statement on p 22, line 16-18:

"Here, we use coincident observations of alkyl nitrates during these elevated O3 periods to estimate the contribution of the light alkanes from oil and gas emissions to O3 production"

The statement in the abstract is an accurate description of what the nitrate analysis shows. The second statement, from p22, is not supported by the discussion in the remainder of this section.

The MS and data establish well that the ozone spikes on 8/8 and 18 are due to upslope events where the O3 spike is nicely correlated with a spike in alkanes and nitrates. This makes sense, since the alkanes are a source of the O3, and as they MS argues, nitrates should be correlated with O3 given the common chemical derivations.

However, NOTHING about this analysis proves that all or even the great majority of the ozone spike comes from alkanes, or from OG! Indeed, this is acknowledged in several places in this discussion.

Because the nitrate signals are very clean spikes from a clean, low background during upslopes, and they are a great fingerprint for upslope ozone generation, the analysis presented here is a great way to quantify the size of the ozone spike that is directly attributable to the upslopes. (Since the ozone trace itself has far more structure, background, variability, local influence, etc., one can't get a clean upslope signal directly from the ozone trace). For this reason, the nitrate analysis well supports the statement in the abstract.

**The authors need to modify or remove the statement above from p 22 and all other statements that state or suggest that the nitrate analysis can ascribe a specific [O3] to light alkanes from OG or emissions from OG. (Statements similar to each of these are currently in the text.)**

In part because there are these variations in the statements about the meaning of the nitrate analysis, this section (p 22 -24) also requires significant reorganization and editing. I believe a lot of it can be cut. Some examples of material that seem extraneous is p 23, lines 13-21 and p 21 line 21 thru p 22 line 3. Moreover, the discussion just needs to be cleaned up. For example, plotting O3 vs nitrate during periods of high O3 is mentioned at the end of p 23, on p 24 lines 18-21, and again on p 25 lines 18-21. This is repetitive. There are other passages which are repetitive or contain extraneous arguments.

P23, line 11 - replace "figure 13" with "figure 12"

P25, line 21 - "and the intercept value was subtracted." Based on the previous discussion of the y-intercept and slope of the O3 vs nitrate plots, I presume this means the intercept value of

ozone. But this (~40 ppb) is not subtracted away from the [nitrate]*slope product.
* * *
**ISSUE 2. Description of source attribution for 8.8 and 8.18 events**

This issue is encapsulated by p 21, line 14 - "dominated" by oil and gas. This is a very strong word. What data supports this? The i/n pentane ratio is 0.97, barely below 1, and a good bit higher than values measured at PAO. (the ratio during the 8/2 event is only slightly higher - it does not seem appropriate to draw such a bright line at 1.00...) Fig 3 clearly shows C2Cl4 peaks on those days, and a C2H2 peak on the 18th. Finally, theOHR data shown in Fig 11 shows very large OVOC contribution to OHR. Considering all of this, it does not seem to me that this description is apt. It seems to be an oversimplification to simply describe this as an OG air mass, as it appears to be heavily influenced by OG, but also distinctly influenced by urban pollution, as lines 20-22 (same page) describe.

Later on, the airmass age analysis (pp 29-30) nicely supports the idea that the airmasses on the 8th and the 18th rapidly moved thru the OG area, picking up OG emissions and then moving up to ROMO. However, its conceivable that they did so after being affected by urban emissions - a variation of the 'regional mixing' scenario described on p 21 lines 20-22. If the authors are relying on photochemical age to show that OG "dominates" for the events on the 8th and 18th, that needs to be made more clear. At the same time, the OHR data doesn't support the "dominated" claim, unless OG contributes a large portion of OVOC (*which if true, is not discussed in the MS*).

Simply put, the MS needs to address these 3 issues - especially the peaks in non-OG VOC and the contribution of OVOC to OHR - during these two upslope events. If they are evidence of regional mixing, statements such as p 21 line 14, and the statement in the abstract that these events are "largely associated with OG VOC" should be made less strong. Alternatively, if the 3 factors I mentioned are not indications of substantial regional mixing, that should be explained, since I think other readers will interpret the data similarly to how I have.
* * *
**SUGGESTION** Page 17 / Fig 6... "At their peak values, NMHC mixing ratios observed at ROMO can be of comparable magnitude to urban/industrial regions" Except, that's not true for some of the VOC shown. It would be very helpful to add a brief discussion of why not. I believe that this reflects the influence of OG VOC on ROMO: ethane, propane, n-but, aromatics. That's why i-but and i-pen are low. But why is n-pen so low at ROMO? why is benzene so high???? why is c2h2 so high???

I realize that this a qualitative, basically contextual result - "can be as high as VOC in urban regions" ... which obviously vary at ton. But still, the different stories for different species are interesting, and rather confusing for readers not familiar with patterns in ambient speciated voc. I suggest adding a couple sentences to tell the reader what's going on here with the various species shown in Fig 6 (at least, some of them).

[Some of this is discussed in the document the authors supplied repsonding to reviewers comments; I believe a few sentences here on the variation between species would be helpful.]

---

## Author Response (AR2)

We appreciate the constructive comments on our manuscript provided by the editor and reviewer. We have taken these into consideration and modified the paper in response. The responses are explained in our response to reviewer comments below with all of our responses are in red type. A marked copy of the manuscript follows the comments/responses.

Comments from Editor:

Pg 2 line 20: Identify Boulder Atmospheric Observatory as BAO since this abbreviation is used later in the paper.

We have now defined BAO where it first appears in the text.

Sect: 2.2.1: How long was needed to each canister collected? Is sampling near the road expected to affect the measurements?

We have clarified this section to address the length of sampling (grab samples – 30-60 s) and possible road emissions.

Sect 2.3.1: Please include model number for 2B Technologies O3 analyzers.

We have added that the POMS3 is a Model 202.

Figure 8: Please add a vertical line separating the in-park canister data from the out of park data

This is a great suggestion and we have made this change.

Pg 12 line 1: Please specify the specific compounds that have the elevated mixing ratios.

We have added that there are elevated mixing ratios of many VOCs including alkanes, alkyl nitrates, and others depending on the event.

Sect 4.2: It seems that OHR should be referred to as VOC reactivity. This would be more consistent with the legend in Fig. 11.

We have made changes in Section 4.2 and 4.3 to be more consistent in our terminology regarding OHR and VOC reactivty.

Pg 15 line 18: Also "as noted in Rosen et al (2004)" should probably be changed to Perring et al (2013)

Thank you for catching this error, we have made this change.

Supplement: Please be consistent with your figure labeling (currently 1S, 2S but S3).

We have revised the numbering in the supplement to be consistent.

Comments From Reviewer:

**ISSUE 1. Nitrate analysis.** The analysis described on pp 22-26 has substantially more explanation, but the discussion is difficult to follow in various places different conclusions are attributed to this analysis. In the abstract, the following statement is made: "For the high ozone events most associated with emissions from oil and gas activities, we estimate that VOCs and NOx from sources along the Front Range contributed ~20 ppbv of additional ozone."
This is quite different than the statement on p 22, line 16-18:
"Here, we use coincident observations of alkyl nitrates during these elevated O3 periods to estimate the contribution of the light alkanes from oil and gas emissions to O3 production"

The statement in the abstract is an accurate description of what the nitrate analysis shows. The second statement, from p22, is not supported by the discussion in the remainder of this section. The MS and data establish well that the ozone spikes on 8/8 and 18 are due to upslope events where the O3 spike is nicely correlated with a spike in alkanes and nitrates. This makes sense, since the alkanes are a source of the O3, and as they MS argues, nitrates should be correlated with O3 given the common chemical derivations. However, NOTHING about this analysis proves that all or even the great majority of the ozone spike comes from alkanes, or from OG! Indeed, this is acknowledged in several places in this discussion. Because the nitrate signals are very clean spikes from a clean, low background during upslopes, and they are a great fingerprint for upslope ozone genera3on, the analysis presented here is a great way to quantify the size of the ozone spike that is directly attributable to the upslopes. (Since the ozone trace itself has far more structure, background, variability, local influence, etc., one can't get a clean upslope signal directly from the ozone trace). For this reason, the nitrate analysis well supports the statement in the abstract.
**The authors need to modify or remove the statement above from p 22 and all other statements that state or suggest that the nitrate analysis can ascribe a specific [O3] to light alkanes from OG or emissions from OG. (Statements similar to each of these are currently in the text.)**
In part because there are these variations in the statements about the meaning of the nitrate analysis, this section (p 22 -24) also requires significant reorganization and editing. I believe a lot of it can be cut. Some examples of material that seem extraneous is p 23, lines 13-21 and p 21 line 21 thru p 22 line 3. Moreover, the discussion just needs to be cleaned up. For example, plotting O3 vs nitrate during periods of high O3 is mentioned at the end of p 23, on p 24 lines 18-21, and again on p 25 lines 18-21. This is repetitive. There are other passages which are repetitive or contain extraneous arguments.

We appreciate the time and effort this reviewer had put into commenting on our manuscript. We have shortened and clarified the alkyl nitrate analysis (issue 1) per the reviewer's suggestions. In addition we made our statements regarding the attribution of additional ozone more consistent to recognize that both oil and gas and other Front Range sources contribute. The high concentrations of alkanes suggest oil and gas contribute more to the additional O3 than other Front Range sources but there is not a clear way to separate the different sources.

P23, line 11 - replace "figure 13" with "figure 12"

We have made this change.

P25, line 21 - "and the intercept value was subtracted." Based on the previous discussion of the
y-intercept and slope of the O3 vs nitrate plots, I presume this means the intercept value of
ozone. But this (~40 ppb) is not subtracted away from the [nitrate]*slope product.

We have corrected this error in the text.

**ISSUE 2. Description of source attribution for 8.8 and 8.18 events**

This issue is encapsulated by p 21, line 14 - "dominated" by oil and gas. This is a very strong word. What data supports this? The i/n pentane ratio
is 0.97, barely below 1, and a good bit higher than values measured at PAO. (the ratio during the 8/2 event is only slightly higher – it does not
seem appropriate to draw such a bright line at 1.00...) Fig 3 clearly shows C2Cl4 peaks on those days, and a C2H2 peak on the 18th. Finally, the
OHR data shown in Fig 11 shows very large OVOC contribution to OHR. Considering all of this, it does not seem to me that this description is apt.
It seems to be an oversimplification to simply describe this as an OG air mass, as it appears to be heavily influenced by OG, but also dis3nctly
influenced by urban pollu3on, as lines 20-22 (same page) describe.

Later on, the airmass age analysis (pp 29-30) nicely supports the idea that the airmasses on the 8th and the 18th rapidly moved thru the OG area,
picking up OG emissions and then moving up to ROMO. However, its conceivable that they did so after being affected by urban emissions – a
variation of the 'regional mixing' scenario described on p 21 lines 20-22. If the authors are relying on photochemical age to show that OG
"dominates" for the events on the 8th and 18th, that needs to be made more clear. At the same 3me, the OHR data doesn't support the
"dominated" claim, unless OG contributes a large portion of OVOC (*which if true, is not discussed in the MS*).

Simply put, the MS needs to address these 3 issues - especially the peaks in non-OG VOC and the contribution of OVOC to OHR - during these two
upslope events. If they are evidence of regional mixing, statements such as p 21 line 14, and the statement in the abstract that these events are
"largely associated with OG VOC" should be made less strong. Alternatively, if the 3 factors I mentioned are not indica3ons of substan3al regional
mixing, that should be explained, since I think other readers will interpret the data similarly to how I have.

While we appreciate the reviewer's concern about the use of "dominated", the data show, both by i- to n-pentane ratio and alkane concentration
relative to Denver observations, that these air masses have a significant oil and gas signature. The upslope events on 8/8 & 8/18 have identical
averages for enhancements above urban for the alkanes (ethane, propane, i&n-butane, i&n-pentane, n-hexane) - both were 3.8 times higher than
the Denver samples. For the mixed events (7/22, 8/11, 8/12), the results were mixed, ranging from 1.0-2.2 and the average was 4.5.  7/22 was
the highest (most O&G influence) at 2.2; 8/11 was intermediate at 1.3, and 8/12 was 1.0.  We have added this information to the text of the
manuscript to better explain the significant oil and gas influence we are describing as dominant.  Even though some urban tracers are observed
in these (8/8 and 8/18) air masses and there are likely other sources mixed in, these air masses look most similar to those found areas near oil
and gas operations. Regarding the pentane ratio, generally it is observed that OG production regions have the lowest ratios and these are usually
below 1. In this case we are using the pentane ratio as a relative measure of oil and gas influence with a lower ratio indicating more oil and gas.
The pentane ratios on 8/11 and 8/12 are above 1.3 indicating a difference in the air masses on 8/8 and 8/18 when the ratio was below 1. Due to
reviewer's suggestions in the alkyl nitrate section we have already change much what we said was OG to a mix of OG and urban. We have made
similar changes here but the data support that 8/8 and 8/18 were more impacted by OG than the other upslope events.

\*\*\*

**SUGGESTION** Page 17 / Fig 6... "At their peak values, NMHC mixing ratios observed at ROMO can be of comparable magnitude to
urban/industrial regions" Except, that's not true for some of the VOC shown. It would be very helpful to add a brief discussion of why not. I believe
that this reflects the influence of OG VOC on ROMO: ethane, propane, n-but, aromatics. That's why i-but and i-pen are low. But why is n-pen so
low at ROMO? why is benzene so high???? why is c2h2 so high???

I realize that this a qualitative, basically contextual result - "can be as high as VOC in urban regions" ... which obviously vary at ton. But still, the
different stories for different species are interes3ng, and rather confusing for readers not familiar with patterns in ambient speciated voc. I suggest
adding a couple sentences to tell the reader what's going on here with the various species shown in Fig 6 (at least, some of them). [Some of this
is discussed in the document the authors supplied responding to reviewers comments; I believe a few sentences here on the variation between
species would be helpful.]

In checking the data we found that the i- and n- species were switched for the Baker (cities) data.  We have fixed that error in the plot and have

[revised manuscript text omitted]

---

## Author Response (AR3)

We have made the two minor corrections as requested by the co-editor. They are marked in red.

[revised manuscript text omitted]